# A Topological Filter for Learning with Label Noise

**Pengxiang Wu**[1*]**, Songzhu Zheng**[2*]**, Mayank Goswami**[3]**, Dimitris Metaxas**[1]**, Chao Chen**[2]

[1]Rutgers University, {pw241,dnm}@cs.rutgers.edu
[2]Stony Brook University, {zheng.songzhu,chao.chen.1}@stonybrook.edu
[3]City University of New York, mayank.isi@gmail.com

## Abstract

Noisy labels can impair the performance of deep neural networks. To tackle this problem, in this paper, we propose a new method for filtering label noise. Unlike most existing methods relying on the posterior probability of a noisy classifier, we focus on the much richer spatial behavior of data in the latent representational space. By leveraging the high-order topological information of data, we are able to collect most of the clean data and train a high-quality model. Theoretically we prove that this topological approach is guaranteed to collect the clean data with high probability. Empirical results show that our method outperforms the state-of-the-arts and is robust to a broad spectrum of noise types and levels.

## 1 Introduction

Corrupted labels are ubiquitous in real world data, and can severely impair the performance of deep neural networks with strong memorization ability [30, 12, 51]. Label noise may arise in mistakes of human annotators or automatic label extraction tools, such as crowd sourcing and web crawling for images [48, 42]. Improving the robustness of deep neural networks to label corruption is critical in many applications [29, 45], yet still remains a challenging problem and largely under-studied.

To combat label noise, state-of-the-art methods often segregate the *clean data* (i.e., samples with uncorrupted labels) from the *noisy* ones. These methods collect clean data iteratively and eventually train a high-quality model. The major challenge is to ensure that the data selection procedure is (1) careful enough to not accumulate errors; and (2) aggressive enough to collect sufficient clean data to train a strong model. Existing methods under this category [27, 21, 16, 43, 31] typically select clean data based on the prediction of the noisy classifier. It is generally assumed that if the noisy classifiers have strong and consistent confidence on a particular label, this label is likely true. However, most of these heuristics do not have a theoretical foundation and thus are not guaranteed to generalize to unseen datasets or noise patterns.

In this paper, we propose to investigate the problem in a novel *topological* perspective. We stipulate that while a noisy classifier's prediction is useful, its latent space representation of the data also contains rich information and should be exploited. Our method is motivated by the following observation: given an ideal feature representation, the clean data are clustered together while the corrupted data are spread out and isolated. This intuition is illustrated in Figure 1(a). We show the spatial distribution pattern of a corrupted dataset with an ideal representation, i.e., the penultimate layer activation (the layer before softmax) of a neural net trained on the original uncorrupted dataset. As is shown in Figure 1(a)(left), the data are well separated into clusters, corresponding to their true labels. Meanwhile, noisy-labeled data (colorful crumbs sprinkled on each cluster) are surrounded by uncorrupted data and thus are isolated.

The above observation inspires us to utilize the spatial topological pattern for label noise filtering. We propose a new method, *TopoFilter*, that collects clean data by selecting the largest connected

---

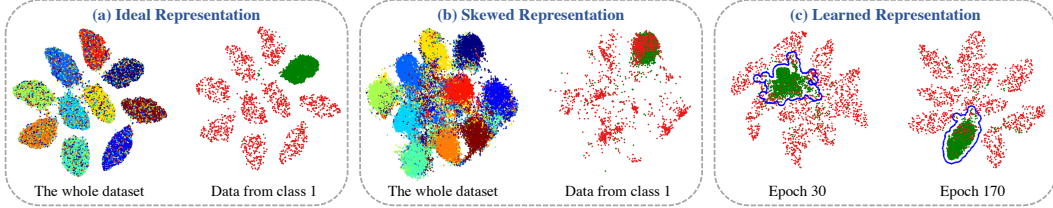

Figure 1: Different representations of a 40% uniformly corrupted CIFAR-10 dataset (visualized using t-SNE). (a) The ideal feature representation (trained on a clean dataset). On the left, we show the whole dataset. Colors correspond to different noisy labels. On the right, we draw all data with label 1. Green points are clean data. Red points are data with corrupted labels. (b) A skewed representation of a noisy classifier, namely, one trained using the corrupted dataset. (c) The learned representations by our algorithm. We show the data of label 1 using the continuously improved representations. The collected data by our method are highlighted with the blue contour.

component of each class and dropping isolated data. Our method leverages the group behavior of data in the latent representation, which has been neglected by previous classifier-confidence-dependent approaches. The challenge is that the ideal representation is unavailable in practice. Training on noisy data leads to a skewed representation (Fig. 1(b)); and the topological intuition does not seem to hold.

To address this issue, we propose an algorithm that uses the topological intuition even with the "imperfect" representation. Our algorithm essentially "peels" the outer most layer of the largest component so that only the core of the component is kept. One particular strength of our method is that it is *theoretically guaranteed to be correct*. We prove (1) *purity*: the collected data have a high chance to be uncorrupted; and (2) *abundancy*: the algorithm can collect a majority of the clean data. These two guarantees ensure the algorithm can collect clean data both carefully and aggressively. Our proof imposes weak assumptions on the representation: (1) the density of the data has a compact support, (2) the true conditional distributions of different labels are continuous, and (3) the decision region of each class of the Bayes optimal classifier is connected. These relative weak assumptions ensures that the theorem still holds on the skewed representation (from a noisy classifier).

We wrap our data collection algorithm to jointly learn the representation and select clean data. To learn the representation, we train a deep net classifier only using the collected clean data. As the classifier continuously improves, it further facilitates the data collection and finally converges to a strong one, as illustrated in Fig. 1(c). We empirically validate the proposed method on different datasets such as CIFAR-10, CIFAR-100 and Clothing1M [47]. Our method consistently outperforms the existing methods under a wide range of noise types and levels.

To summarize, we propose the first theoretically guaranteed algorithm for label noise that exploits a topological view of the noisy data representation. Our paper offers both the algorithmic intuition and the theoretical rationale on how spatial pattern and group behavior of data in the latent space can be informative of the model training. We believe the geometry and topology of data in the latent space should be further explored for better understanding and regulating of neural networks.

**Related works.** One representative class of methods for handling label noise aim to improve the robustness by modeling the noise transition process [38, 33, 15, 18]. However, the estimation of noise transformation is non-trivial, and these methods generally require additional access to the true labels or depend on strong assumptions, which could be impractical. In contrast to these works, our method does not rely on noise modeling, and is thus more generic and flexible.

A number of approaches have sought to develop noise-robust loss to help resist label corruption. One typical idea is to reduce the influence of noisy samples with carefully designed losses [35, 1, 52, 40, 44, 25, 14, 6] or regularization terms [20, 28, 23]. Closely related to this philosophy, other approaches focus on adaptively re-weighting the contributions of the noisy samples to the loss. The re-weighting functions could be pre-specified based on heuristics [5, 43] or learned automatically [21, 36, 37]. Our method is independent of the loss function, and can be combined with any of them.

Another direction seeks to improve the label quality by correcting the noisy labels to the underlying true ones [47, 41, 42, 24, 39, 49]. To predict the true labels, these approaches generally require additional clean labels, complex interventions into the learning process, or an expensive detection process for noise model estimation. Moreover, these methods are based on heuristics without

theoretical guarantees, and tend to be sensitive to the hyper-parameters (e.g., learning rate and loss coefficients). Zheng et al. [54] showed that assuming the noisy classifier approximates the noisy conditional distribution well everywhere, the noisy classifier can help correct labels with high probability. Our theorem has a weaker assumption on the noisy classifier's quality.

Our work can be categorized as a data-selection method. Some methods choose the clean data based on the prediction agreements among different networks [27, 31]. Others train the networks only on samples with small losses and exchange the error flows between networks [16, 10, 50]. These methods typically train multiple networks, and is thus computationally expensive and hard to tune. The data-selection process in these methods is generally based on heuristics without guarantees.

A few existing works also seek to handle the label noise by probing the spatial properties of data. Wang *et al.* [43] propose to detect noisy data using spatial outlier detection. Gao *et al.* [13] use $k$-nearest neighbor to correct noisy labels. Both of these methods rely on local spatial information. They fail to explore global structural information that could reveal critical common patterns, such as topology. Lee *et al.* [22] model the spatial distributions with a generative model and train a robust generative classifier using all noisy data. For completeness, we also refer to works studying KNN-induced connectivity [26, 7], which only focus on the unsupervised setting.

## 2 Method

Our algorithm jointly trains a neural network and collects clean data. At each epoch, clean data are collected based on the their spatial topology in the latent space of the current network. Meanwhile, only clean data are used to further train the network. In the beginning, we use an early-stopped noisy classifier to learn the representation. It has been observed that an early-stopped model will learn meaningful feature without overfitting the noise [51, 2]. Such a network, although not powerful enough, can provide a reasonable initial representation for our data-collection algorithm to start.

Below we present our algorithm. We first provide a baseline, called *TopoCC*. It collects clean data only using the largest connected component. However, this is insufficient due to the imperfect representation. Next, we present our main algorithm, called *TopoFilter*, that further "peels" the largest component and only keeps its core.

Our algorithm for data selection is as follows. Let $v$ be the input data and $x$ to be the latent feature given by network by taking input $v$, we probe the spatial data distributions by building a $k$-nearest neighbor (KNN) graph $G$ upon $x$. From $G$ we further derive the subgraph $G_i$ for class $i$ by removing the vertices belonging to other classes and their associated edges. On each $G_i$, we find the largest connected component $Q_i$ and consider the data belonging to $Q_i$ as clean. Eventually we have a collection of potentially clean data $C = \cup_i Q_i$. Intuitively, the clean data will be regularly and densely distributed in the feature space. They will form a salient topological structure (connected component), which could thus be captured by the algorithm. Plugging this data-collection procedure into our joint training algorithm gives the baseline *TopoCC*.

However, simply relying on connected components is insufficient; the geometry and thus connectivity of the data is not fully reliable due to the imperfect representation. In particular, near the outer most layer of the largest connected component, the data can easily be corrupted. We need to remove these data in order to improve the purity of the selected data. In particular, for a given sample $x$ belonging to one of the largest connected components $S_i$, with label $\widetilde{y}$, find its $k$-nearest neighbors $KNN(x)$ from $S$ (the union of largest components for each class). Then we consider $x$ as clean if at least a fraction $\zeta$ of its neighbors have the same label $\widetilde{y}$. As is illustrated in Section 2.1 and Section 3, this additional filtering of the largest component, called the $\zeta$-filtering, is essential to the success of our method. We name this method *TopoFilter*. Details are in Algorithm 1. In practice, we observe that our algorithm is insensitive to the choice of $\zeta$. More empirical study can be found in the supplementary material. At the end of this section, we will provide details on how to choose $\zeta$ based on the theory.

### 2.1 Theoretical Guarantee of the Algorithm

Next we provide a detailed analysis of our method. We show that after running our algorithm once, the collected data are *pure*, i.e., have high probability to be clean (Theorem 1). Meanwhile, we prove that the algorithm collects *abundant* clean data, i.e., a sufficiently large amount of clean data (Theorem 2). Both theorems are critical in ensuring we train a high-quality model despite the label noise. Note that our theoretical results are one-shot. We leave the convergence result as future work.

---

**Algorithm 1** TopoFilter

---

1: **Input:** Noisy training data $\mathcal{S}$, milestone $m$, training epochs $N$, number of classes $\Gamma$, number of neighbors $k$, filtering parameter $\zeta$
2: **Output:** Collected clean data $C$
3: Initialize $C \leftarrow \emptyset, \widehat{\mathcal{S}} \leftarrow \mathcal{S}$
4: **for** $t = 1, \cdots, N$ **do**
5:      Train network on $\widehat{\mathcal{S}}$
6:      **if** $t \geq m$ **then**
7:          Extract feature vectors $\boldsymbol{x}$ from training data $\mathcal{S}$
8:          Compute $k$-NN graph $G$ over $\boldsymbol{x}$
9:          **for** $i = 1, \cdots, \Gamma$ **do**
10:              Construct subgraph $G_i$ by selecting feature vectors $\boldsymbol{x}^{(i)}$ from $i$-th class and removing all edges associated with $\boldsymbol{x}^{(j)}$ for $j \neq i$
11:              Compute the largest connected component $Q_i$ over $G_i$
12:              $C \leftarrow C \cup Q_i$
13:          **end for**
14:          Find outliers $O$ within $C$ based on $\zeta$-filtering; update $C \leftarrow C \backslash O$
15:          $\widehat{\mathcal{S}} \leftarrow C$
16:      **end if**
17: **end for**

---

We first introduce notations. Next, we present the purity and abundancy theorems respectively. Due to space constraints, we mainly present the theorems and their intuitions. Details of the proofs can be found in the supplemental material.

**Notations.** We focus on binary classification. Assume that the data points and labels lie in $\mathcal{X} \times \mathcal{Y}$, where the feature space $\mathcal{X} \subset \mathbb{R}^d$ and label space $\mathcal{Y} = \{0, 1\}$. A datum $x$ and its true label $y$ follow a distribution $F \sim \mathcal{X} \times \mathcal{Y}$. Let $f(\boldsymbol{x}) := \sum_{i \in \{0,1\}} F(\boldsymbol{x}, i)$ be the density at $\boldsymbol{x}$. Denote by $\tilde{y}$ the observed (potentially noisy) label. Due to label noise, label $y = i$ is flipped to $\tilde{y} = j$ with probability $\tau_{ij}$ and is assumed to be independent of $\boldsymbol{x}$.

Let $\mathbf{X} \subset \mathcal{X}$ be the finite set of features in the data sample, and let $G(\mathbf{X}, k)$ be the mutual $k$-nearest neighbor graph on $\mathbf{X}$ using the Euclidean metric on $\mathcal{X}$, whose edge set $E = \{(\boldsymbol{x}_1, \boldsymbol{x}_2) \in \mathbf{X} \times \mathbf{X} \mid \boldsymbol{x}_1 \in KNN(\boldsymbol{x}_2) \text{ or } \boldsymbol{x}_2 \in KNN(\boldsymbol{x}_1)\}$. Also, $\forall i \in \{0, 1\}$, let $G_i(\mathbf{X}, k)$ be the induced subgraph of $G(\mathbf{X}, k)$ consisting only of vertices $\boldsymbol{x} \in \mathbf{X}$ with label $\widetilde{y}(\boldsymbol{x}) = i$.

Let $\eta_i(\boldsymbol{x}) = P(y = i \mid \boldsymbol{x})$ and $\widetilde{\eta}_i(\boldsymbol{x}) = P(\widetilde{y} = i \mid \boldsymbol{x})$ be the conditional probability of the clean and noisy labels given a feature $\boldsymbol{x}$, respectively. Since this is binary setting, we have $\eta_i(\boldsymbol{x}) = 1 - \eta_{1-i}(\boldsymbol{x})$ and $\widetilde{\eta}_i(\boldsymbol{x}) = 1 - \widetilde{\eta}_{1-i}(\boldsymbol{x})$. Since $\widetilde{\eta}_i(\boldsymbol{x}) = \tau_{1-i,i}\eta_i(\boldsymbol{x})$, these probabilities satisfy a linear relationship. $\widetilde{\eta}_i(\boldsymbol{x}) = (1 - \tau_{01} - \tau_{10})\eta_i(\boldsymbol{x}) + \tau_{1-i,i}, \forall i \in \{0, 1\}$. Define the superlevel set $L(t) = \{\boldsymbol{x} \mid \max(\eta_1(\boldsymbol{x}), \eta_0(\boldsymbol{x})) \geq t\}$, and let $\mu(L(t))$ be the probability measure of $L(t)$. Lastly, the indicator function $I_A(\boldsymbol{x}) = 1$ if $\boldsymbol{x} \in A$ and $I_A(\boldsymbol{x}) = 0$ otherwise.

Consider an algorithm $\mathcal{A}$ that takes as input a random sample of size $n$, $S_n = \{(\boldsymbol{x}_i, \tilde{y}(\boldsymbol{x}_i))\}_{i=1}^n$. The set of features of the data is $\mathbf{X} = \{\boldsymbol{x}_i\}_{i=1}^n \subset \mathcal{X}$. Algorithm $\mathcal{A}$ then outputs $\cup_{i \in \{0,1\}} C_i$, where $C_i \subseteq \mathbf{X}_i := \{\boldsymbol{x} : \tilde{y}(\boldsymbol{x}) = i\}$ is the claimed "clean" set for label $i$.

**Definition 1.** *(Purity) We define two kinds of purity of $\mathcal{A}$ on $S_n$. One captures the worst-case behavior of the algorithm, while the other captures the average-case behavior.*

**1. Minimum Purity** $\ell_{S_n, \mathcal{A}} := \min_{i \in \{0,1\}} \min_{\boldsymbol{x} \in C_i} P(y = i \mid \widetilde{y} = i, \boldsymbol{x}) = \min_{i \in \{0,1\}} \min_{\boldsymbol{x} \in C_i} \tau_{ii} \frac{\eta_i(\boldsymbol{x})}{\widetilde{\eta}_i(\boldsymbol{x})}$.

**2. Average Purity** $\ell'_{S_n, \mathcal{A}} := \sum_{i \in \{0,1\}} \mathbb{E}_{\boldsymbol{x} \in C_i}[P(y = i \mid \widetilde{y} = i, \boldsymbol{x})] = \sum_{i \in \{0,1\}} \frac{1}{|C_i|} \sum_{\boldsymbol{x} \in C_i} \tau_{ii} \frac{\eta_i(\boldsymbol{x})}{\widetilde{\eta}_i(\boldsymbol{x})}$.

We define the following three sets, which form a partition of $\mathcal{X}$:

$$
\begin{aligned}
A_i^+ &= \left\{ \boldsymbol{x} : \widetilde{\eta}_i(\boldsymbol{x}) > \max\left(\tfrac{1}{2}, \tfrac{1+\tau_{i,1-i}-\tau_{1-i,i}}{2}\right) \right\} = \left\{ \boldsymbol{x} : \eta_i(\boldsymbol{x}) > \max\left(\tfrac{1}{2}, \tfrac{1/2 - \max(\tau_{10}, \tau_{01})}{2(1-\tau_{10}-\tau_{01})}\right) \right\}, \\
A_i^- &= \left\{ \boldsymbol{x} : \widetilde{\eta}_i(\boldsymbol{x}) < \min\left(\tfrac{1}{2}, \tfrac{1+\tau_{i,1-i}-\tau_{1-i,i}}{2}\right) \right\} = \left\{ \boldsymbol{x} : \eta_i(\boldsymbol{x}) < \min\left(\tfrac{1}{2}, \tfrac{1/2 - \max(\tau_{10}, \tau_{01})}{2(1-\tau_{10}-\tau_{01})}\right) \right\}, \\
A^b &= \mathcal{X} \setminus (A_i^+ \cup A_i^-).
\end{aligned}
$$

$A_i^+$ is the *good* region where clean Bayes classifier and noisy Bayes classifier have the same correct prediction $i$. Notice that $A_i^+ = A_{1-i}^-$. The whole idea of our algorithm is to collect as many points in $A_i^+$ as we can in region $A_i^+$ because those points are most likely to be clean. Meanwhile, the algorithm throws away uncertain points in the region $A^b$ whose points are near to the decision boundary of Bayes classifier and are more likely to be corrupted.

**Assumptions.** To establish our theorems, we will assume the following reasonable conditions:

**A1:** $f(\boldsymbol{x})$ (the density on the feature space) has compact support.

**A2:** $\forall i \in \{0, 1\}$, $\eta_i(\boldsymbol{x})$ is continuous.[1]

**A3:** $\forall i \in \{0, 1\}$, $A_i^+$ is a connected set.

**A4:** $\tau_{10}, \tau_{01} \in \left(0, \frac{1}{2}\right)$.

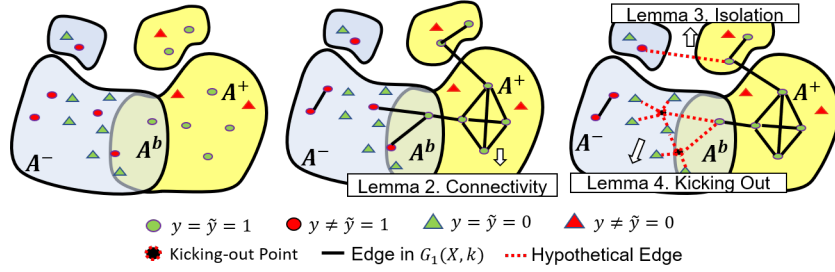

Figure 2: Algorithm illustration. Points from $A_i^+$ are all connected; points in $A_i^- \cup A^b$ are kicked out.

Denote by $\mathcal{A}_0$ the naive algorithm which takes input $S_n$ and simply outputs $C_i = \mathbf{X}_i$ for $i = 0, 1$, i.e., does no processing and treats corrupted labels as clean. The purity of $\mathcal{A}_0$ is the "default" purity of the data set. Denote our algorithm with parameter $\zeta$ by $\mathcal{A}_\zeta$. Let $e$ be the natural constant.

**Theorem 1.** *(Purity Guarantee)* $\forall \delta > 0$, $\forall \zeta > \frac{1+|\tau_{10}-\tau_{01}|}{2}$, *there exist* $N(\delta, \zeta) > 0$, $c_1(\zeta) > 0$ , $c_2 \in \left(0, \frac{e-1}{e}\right)$, *and an increasing function* $g_1(\zeta) \in \left[\frac{[(2\zeta+1+|\tau_{10}-\tau_{01}|)-4\max(\tau_{10},\tau_{01})]\min(\tau_{11},\tau_{00})}{(2\zeta+1+|\tau_{10}-\tau_{01}|)(1-\tau_{10}-\tau_{01})}, 1\right]$ *and function* $g_2(\zeta) > 0$, *such that* $\forall n \geq N(\delta, \zeta)$, $\forall q > 1$ *and* $\forall k \in [c_1(\zeta)\log^q n, c_2 n]$:

*1.* $P\left[(\ell_{S_n, \mathcal{A}_\zeta} - \ell_{S_n, \mathcal{A}_0}) > g_1(\zeta)\right] \geq 1 - \delta$, *and*

*2.* $P\left[(\ell'_{S_n, \mathcal{A}_\zeta} - \ell'_{S_n, \mathcal{A}_0}) > g_2(\zeta)\right] \geq 1 - \delta$.

**Sketch of Proof.** The complete proof is in the supplementary materials; here we provide a sketch and the main lemmas used to prove Theorem 1. Firstly, we show that for a given $t \in [0, 1)$, when the sample size $n$ is large enough and the number of neighbors $k$ is set to be $\Omega(\log^q(n))$, then all data points from $X_i(t) := L(t) \cap \mathbf{X}_i$ will be connected in $G_i(\mathbf{X}, k)$.

**Lemma 1.** *(Connectivity).* $\forall \delta > 0$ *and* $\forall t \in [0, 1)$, *there exist* $N(\delta, t) > 0$, *and* $c_1(t) > 0$ *such that* $\forall n \geq N(\delta, t)$, $\forall q > 1$, $\forall k > c_1(t)\log^q(n)$ *and* $\forall i \in \{0, 1\}$, $X_i(t)$ *is connected in* $G_i(\mathbf{X}, k)$ *with probability at least* $1 - \delta$.

Let $\zeta' = \frac{1}{2}\left(\zeta + \frac{1+|\tau_{10}-\tau_{01}|}{2}\right)$. Notice that because $\zeta > \frac{1+|\tau_{10}-\tau_{01}|}{2}$, $\zeta > \zeta'$ and $L(\zeta) \subset L(\zeta') \subset A_i^+$. Next we prove that when $k$ is not too large, there will be no points in $A_i^+ \cap L(\zeta)$ that have an edge to a point in $L(\zeta')^c$, where $L(\zeta')^c$ is the complement of $L(\zeta')$. Let $\overline{L(\zeta')^c}$ be the closure of $L(\zeta')^c$, and denote $\mathbf{X}_i^c(\zeta') := \overline{L(\zeta')^c} \cap \mathbf{X}_i$. Define $r_0^i = \min \|\boldsymbol{x}_1^i - \boldsymbol{x}_2^i\|$ for $\boldsymbol{x}_1^i \in X_i(\zeta)$ and $\boldsymbol{x}_2^i \in X_i^c(\zeta')$. Also observe that $A_i^+$, $A_i^-$ and $A^b$ form a partition of the domain, which along with the assumption A1 of compact support implies that $r_0 > 0$. Let $V_d$ to be the volume of $d$-dimensional

---

unit ball. Let $p_\zeta^{(i)} = \min\limits_{\boldsymbol{x} \in L(\zeta) \cap A_i^+} f(\boldsymbol{x}) V_d(r_0^i)^d$ and $p_{\zeta'}^{(i)} = \inf\limits_{\boldsymbol{x} \in L(\zeta')^c \cap A_i^{+c}} f(\boldsymbol{x}) V_d(r_0^i)^d$. Since $f(\boldsymbol{x})$ has compact support, $p_\zeta^{(i)} > 0$ and $p_{\zeta'}^{(i)} > 0$.

[1]This is a significantly weaker conditions on $\eta_i$ than is assumed in many prior works on KNN classifiers (such as Hölder continuous [3], Lipschitz continuous [13], etc.). This condition upper bounds the measure of region that is close to the decision boundary of Bayes decision rule. This assumption is also adopted in [7, 4, 34] and is reasonably well accepted.

**Lemma 2. (Isolation).** $\forall \delta > 0$, $\forall \zeta > \frac{1+|\tau_{10}-\tau_{01}|}{2}$, there exists $N(\zeta, \delta) > 0$ and $c_2 \in \left(0, \frac{e-1}{e}\right)$ such that $\forall n \geq N(\zeta, \delta)$, $\forall k < c_2 \min\limits_{i \in \{0,1\}} \min\left(p_\zeta^{(i)}, p_{\zeta'}^{(i)}\right)(n-1) + 1$ and $\forall i \in \{0, 1\}$:

$$P\left(\nexists edge = (u, v) \in G_i(\mathbf{X}, k) : u \in \mathbf{X}_i(\zeta), v \in \mathbf{X}_i^c(\zeta')\right) \geq 1 - \delta.$$

Then we show after the $\zeta$-filtering step, with large probability there will be no points from $L(\zeta')^c$ in our final set. Denote $C^{(i)}(\zeta)$ to be the data of type $i$ finally kept by the algorithm using parameter $\zeta$.

**Lemma 3. ($\zeta$-filtering).** $\forall \delta > 0$ and $\zeta \in \left(\frac{1+|\tau_{10}-\tau_{01}|}{2}, 1\right)$, there exists $N(\zeta, \delta) > 0$ and $c_3(\zeta) > 0$, such that $\forall n \geq N(\zeta, \delta)$, $k > c_3(\zeta) \log(2n/\delta)$ and $\forall i \in \{0, 1\}$ :

$$P\left(C^{(i)}(\zeta) \cap L(\zeta')^c = \emptyset\right) \geq 1 - \delta.$$

To obtain Theorem 1, we combine the above lemmas as follows. The minimum purity of a data point retained by our algorithm is lower bounded by the purity of a point in the level set $\{x : \tilde{\eta}(\boldsymbol{x}) = \zeta'\}$, which followed by algebraic calculation gives us the first part of the theorem. For the second part of the theorem, we need to calculate the average purity for our algorithm, which requires a more involved integral calculation over $L(\zeta')$. The complete proof is in the supplementary materials.

Our next theorem (proof in supplementary materials) gives a lower bound on the number of points that will be eventually kept by our algorithm $\mathcal{A}_\zeta$. As we will see, there will be a trade off between the size of the retained set and its purity. A larger $\zeta$ will result in smaller connected component but higher purity, while small $\zeta$ gives large connected component but lower purity.

**Theorem 2. (Abundancy)** Let $n_c = \#\left\{\bigcup_i C^{(i)}(\zeta)\right\}$. Then $\forall \delta > 0$, $\forall \zeta > \frac{1+|\tau_{10}-\tau_{01}|}{2}$, $\forall \epsilon > 0$, there exist constants $c_1(\zeta) > 0$, $c_2 \in \left(0, \frac{e-1}{e}\right)$ and $N(\delta, \zeta, \epsilon) > 0$, such that $\forall n \geq N(\delta, \zeta, \epsilon)$, and $\forall k \in [c_1(\zeta) \log^q n, c_2 n]$, we have $P\left[|n_c/n - \mu(L(\zeta))| \leq \epsilon\right] \geq 1 - \delta$.

**Remark on choosing $\zeta$:** In the beginning epochs, because of the corruption of the distribution, one does not expect high confidence in the network classifier. In order to guarantee a minimum level of purity, we set $\zeta$ to be high, say $3/4$. While in these rounds the purity is high and the abundancy is lower bounded, we still want to increase abundancy further. In the later epochs, after training on this clean(er) data, we develop more confidence in our network classifier, and hence we reduce $\zeta$, letting $\zeta$ go to $(1/2 + \epsilon)$ for a very small $\epsilon > 0$, which corresponds to collecting data from the boundary region of the Bayesian classifier. More discussion can be found in the supplemental material.

## 3  Experiments

**Synthetic datasets.** We test the proposed method on CIFAR-10 and CIFAR-100, which are popularly used for the study of label noise. We preprocess each image by normalizing it with the training set mean and standard deviation. For each of the datasets, we split 20% from the training set as validation data. The validation set could be noisy or clean, whereas we only use clean testing data. We employ ResNet-18 [17] as the experimental network, which achieves reasonable performance on the two datasets, with 92.0% and 70.4% test accuracies on CIFAR-10 and CIFAR-100, respectively. In the supplementary material, we provide additional results on the ModelNet40 [46] dataset, which offers a domain different from images.

**Generating corrupted labels.** Similar to [33], we artificially corrupt the labels by constructing the noise transition matrix $T$, where $T_{ij} = P(\widetilde{y} = j | y = i) = \tau_{ij}$ defines the probability that a true label $y = i$ is flipped to $j$. Then for each sample with label $i$, we replace its label with the one sampled from the probability distribution given by the $i$-th row of matrix $T$. In this work, we consider two types of noise, both of which can be formulated using the transition matrices. (1) *Uniform flipping*: the true label $i$ is corrupted uniformly to other classes, i.e., $T_{ij} = \tau/(\mathcal{C}-1)$ for $i \neq j$, and $T_{ii} = 1 - \tau$, where $\tau$ is the constant noise level; (2) *Pair flipping*: the true label $i$ is flipped to $j$ or stays unchanged

Table 1: Test accuracies (%) on CIFAR-10 and CIFAR-100 under different noise types and fractions. The average accuracies and standard deviations over 5 trials are reported. We perform unpaired $t$-test (95% significance level) on the difference between the test accuracies, and observe the improvement due to our method over state-of-the-art methods is statistically significant for all noise settings.

| Dataset | Method | Uniform Flipping | | | | Pair Flipping | | |
|---|---|---|---|---|---|---|---|---|
| | | 20% | 40% | 60% | 80% | 20% | 30% | 40% |
| CIFAR-10 | Standard | $85.7 \pm 0.5$ | $81.8 \pm 0.6$ | $73.7 \pm 1.1$ | $42.0 \pm 2.8$ | $88.0 \pm 0.3$ | $86.4 \pm 0.4$ | $84.9 \pm 0.7$ |
| | Forgetting | $86.0 \pm 0.8$ | $82.1 \pm 0.7$ | $75.5 \pm 0.7$ | $41.3 \pm 3.3$ | $89.5 \pm 0.2$ | $88.2 \pm 0.1$ | $85.0 \pm 1.0$ |
| | Bootstrap | $86.4 \pm 0.6$ | $82.5 \pm 0.1$ | $75.2 \pm 0.8$ | $42.1 \pm 3.3$ | $88.8 \pm 0.5$ | $87.5 \pm 0.5$ | $85.1 \pm 0.3$ |
| | Forward | $85.7 \pm 0.4$ | $81.0 \pm 0.4$ | $73.3 \pm 1.1$ | $31.6 \pm 4.0$ | $88.5 \pm 0.4$ | $87.3 \pm 0.2$ | $85.3 \pm 0.6$ |
| | Decoupling | $87.4 \pm 0.3$ | $83.3 \pm 0.4$ | $73.8 \pm 1.0$ | $36.0 \pm 3.2$ | $89.3 \pm 0.3$ | $88.1 \pm 0.4$ | $85.1 \pm 1.0$ |
| | MentorNet | $88.1 \pm 0.3$ | $81.4 \pm 0.5$ | $70.4 \pm 1.1$ | $31.3 \pm 2.9$ | $86.3 \pm 0.4$ | $84.8 \pm 0.3$ | $78.7 \pm 0.4$ |
| | Co-teaching | $89.2 \pm 0.3$ | $86.4 \pm 0.4$ | $79.0 \pm 0.2$ | $22.9 \pm 3.5$ | $90.0 \pm 0.2$ | $88.2 \pm 0.1$ | $78.4 \pm 0.7$ |
| | Co-teaching+ | $89.8 \pm 0.2$ | $86.1 \pm 0.2$ | $74.0 \pm 0.2$ | $17.9 \pm 1.1$ | $89.4 \pm 0.2$ | $87.1 \pm 0.5$ | $71.3 \pm 0.8$ |
| | IterNLD | $87.9 \pm 0.4$ | $83.7 \pm 0.4$ | $74.1 \pm 0.5$ | $38.0 \pm 1.9$ | $89.3 \pm 0.3$ | $88.8 \pm 0.5$ | $85.0 \pm 0.4$ |
| | RoG | $89.2 \pm 0.3$ | $83.5 \pm 0.4$ | $77.9 \pm 0.6$ | $29.1 \pm 1.8$ | $89.6 \pm 0.4$ | $88.4 \pm 0.5$ | $86.2 \pm 0.6$ |
| | PENCIL | $88.2 \pm 0.2$ | $86.6 \pm 0.3$ | $74.3 \pm 0.6$ | $45.3 \pm 1.4$ | $90.2 \pm 0.2$ | $88.3 \pm 0.2$ | $84.5 \pm 0.5$ |
| | GCE | $88.7 \pm 0.3$ | $84.7 \pm 0.4$ | $76.1 \pm 0.3$ | $41.7 \pm 1.0$ | $88.1 \pm 0.3$ | $86.0 \pm 0.4$ | $81.4 \pm 0.6$ |
| | SL | $89.2 \pm 0.5$ | $85.3 \pm 0.7$ | $78.0 \pm 0.3$ | $44.4 \pm 1.1$ | $88.7 \pm 0.3$ | $86.3 \pm 0.1$ | $81.4 \pm 0.7$ |
| | TopoCC | $89.6 \pm 0.3$ | $86.0 \pm 0.5$ | $78.7 \pm 0.5$ | $43.0 \pm 2.0$ | $89.8 \pm 0.3$ | $87.3 \pm 0.3$ | $85.4 \pm 0.4$ |
| | TopoFilter | $\mathbf{90.2 \pm 0.2}$ | $\mathbf{87.2 \pm 0.4}$ | $\mathbf{80.5 \pm 0.4}$ | $\mathbf{45.7 \pm 1.0}$ | $\mathbf{90.5 \pm 0.2}$ | $\mathbf{89.7 \pm 0.3}$ | $\mathbf{87.9 \pm 0.2}$ |
| CIFAR-100 | Standard | $56.5 \pm 0.7$ | $50.4 \pm 0.8$ | $38.7 \pm 1.0$ | $18.4 \pm 0.5$ | $57.3 \pm 0.7$ | $52.2 \pm 0.4$ | $42.3 \pm 0.7$ |
| | Forgetting | $56.5 \pm 0.7$ | $50.6 \pm 0.9$ | $38.7 \pm 1.0$ | $18.4 \pm 0.4$ | $57.5 \pm 1.1$ | $52.4 \pm 0.8$ | $42.4 \pm 0.8$ |
| | Bootstrap | $56.2 \pm 0.5$ | $50.8 \pm 0.6$ | $37.7 \pm 0.8$ | $19.0 \pm 0.6$ | $57.1 \pm 0.9$ | $53.0 \pm 0.9$ | $43.0 \pm 1.0$ |
| | Forward | $56.4 \pm 0.4$ | $49.7 \pm 1.3$ | $38.0 \pm 1.5$ | $12.8 \pm 1.3$ | $56.8 \pm 1.0$ | $52.7 \pm 0.5$ | $42.0 \pm 1.0$ |
| | Decoupling | $57.8 \pm 0.4$ | $49.9 \pm 1.0$ | $37.8 \pm 0.7$ | $17.0 \pm 0.7$ | $60.2 \pm 0.9$ | $54.9 \pm 0.1$ | $47.2 \pm 0.9$ |
| | MentorNet | $62.9 \pm 1.2$ | $52.8 \pm 0.7$ | $36.0 \pm 1.5$ | $15.1 \pm 0.9$ | $62.3 \pm 1.3$ | $55.3 \pm 0.5$ | $44.4 \pm 1.6$ |
| | Co-teaching | $64.8 \pm 0.2$ | $60.3 \pm 0.4$ | $46.8 \pm 0.7$ | $13.3 \pm 2.8$ | $63.6 \pm 0.4$ | $58.3 \pm 1.1$ | $48.9 \pm 0.8$ |
| | Co-teaching+ | $64.2 \pm 0.4$ | $53.1 \pm 0.2$ | $25.3 \pm 0.5$ | $10.1 \pm 1.2$ | $60.9 \pm 0.3$ | $56.8 \pm 0.5$ | $48.6 \pm 0.4$ |
| | IterNLD | $57.9 \pm 0.4$ | $51.2 \pm 0.4$ | $38.1 \pm 0.9$ | $15.5 \pm 0.8$ | $58.1 \pm 0.4$ | $53.0 \pm 0.3$ | $43.5 \pm 0.8$ |
| | RoG | $63.1 \pm 0.3$ | $58.2 \pm 0.5$ | $47.4 \pm 0.8$ | $20.0 \pm 0.9$ | $67.1 \pm 0.6$ | $65.6 \pm 0.4$ | $58.8 \pm 0.1$ |
| | PENCIL | $64.9 \pm 0.3$ | $61.3 \pm 0.4$ | $46.6 \pm 0.7$ | $17.3 \pm 0.8$ | $67.5 \pm 0.5$ | $66.0 \pm 0.4$ | $61.9 \pm 0.4$ |
| | GCE | $63.6 \pm 0.6$ | $59.8 \pm 0.5$ | $46.5 \pm 1.3$ | $17.0 \pm 1.1$ | $64.8 \pm 0.9$ | $61.4 \pm 1.1$ | $50.4 \pm 0.9$ |
| | SL | $62.1 \pm 0.4$ | $55.6 \pm 0.6$ | $42.7 \pm 0.8$ | $19.5 \pm 0.7$ | $59.2 \pm 0.6$ | $55.1 \pm 0.7$ | $44.8 \pm 0.1$ |
| | TopoCC | $64.1 \pm 0.8$ | $57.3 \pm 1.6$ | $45.1 \pm 1.1$ | $19.1 \pm 0.6$ | $66.3 \pm 0.8$ | $62.3 \pm 0.9$ | $58.3 \pm 0.9$ |
| | TopoFilter | $\mathbf{65.6 \pm 0.3}$ | $\mathbf{62.0 \pm 0.6}$ | $\mathbf{47.7 \pm 0.5}$ | $\mathbf{20.7 \pm 1.2}$ | $\mathbf{68.0 \pm 0.3}$ | $\mathbf{66.7 \pm 0.6}$ | $\mathbf{62.4 \pm 0.2}$ |

with probabilities $T_{ij} = \tau$ and $T_{ii} = 1 - \tau$, respectively. Pair flipping is used to simulate the real mistakes made by human labelers on similar classes. For more details we refer the readers to [33].

**Baselines.** We compare the proposed method with the following representative approaches. (1) *Standard*, which is simply the standard deep network trained on noisy datasets; (2) *Forgetting* [2]; (3) *Bootstrap* [35]; (4) *Forward Correction* [33]; (5) *Decoupling* [27]; (6) *MentorNet* [21]; (7) *Co-teaching* [16]; (8) *Co-teaching+* [50]; (9) *IterNLD* [43]; (10) *RoG* [22]; (11) *PENCIL* [49]; (12) *GCE* [52]; (13) *SL* [44]. These methods are from different research directions.

We implement our method with PyTorch[2]. For data selection, we compute the KNN graph with CUDA and calculate the largest connected component in C++: the overall computational cost per iteration is less than 1s on an Intel Xeon Gold 5218 CPU and a single NVIDIA RTX 4000 GPU. We use a batch size of 128 and train the networks for 180 epochs to ensure convergence. We train the network with Adam optimizer using its default parameters. The data selection is performed every 5 epochs. All experiments are randomly repeated 5 times, and the mean and standard deviation values are reported. All the methods use clean validation set for model selection. Note that, our method is robust to the validation set, as shown below.

**Results.** Table 1 shows the performance of different methods. We observe that TopoFilter consistently outperforms the competitive methods across different noise settings. This suggests the benefits of leveraging spatial pattern for label denoising. Notice that, although the posterior probabilities employed by a few works are closely related to the penultimate layer features used in our method, they intrinsically undergo a dimension reduction process and may lose some critical information. This would explain the superior performance of our method to some degree.

From Table 1 we also observe that simply using largest connected components (TopoCC) or spatial outliers (IterNLD) are less effective. This is because the data in the connected components could still

Table 2: Classification accuracy (%) on Clothing1M test set.

| Method | Standard | Forward | D2L | Joint Opt. | PENCIL | MLNT | DY | GCE | SL | **TopoFiler** |
|---|---|---|---|---|---|---|---|---|---|---|
| Accuracy | 68.94 | 69.84 | 69.47 | 72.23 | 73.49 | 73.47 | 71.00 | 69.75 | 71.02 | **74.10** |

contain noise, and thereby hurts the model performance eventually. Similarly, the outlier detection is not reliable as the noisy data could form a small cluster and thus do not appear to be outliers spatially, as illustrated in Fig. 1(a)(b).

**Behavior analysis.** In Fig. 3(a) we show the validation accuracy of TopoFilter on the two datasets with different noise settings. As is shown, the accuracy of TopoFilter does not drop throughout the training process. This indicates that our method filters out the noise successfully and stably. This is further confirmed in Fig. 3(b)-(e), where the collected data pool preserves high purity during training with its size approaching the limit steadily.

**Parameter analysis.** In Fig. 4(a) we show that TopoFilter is robust to the size and purity of validation set. Notably, it achieves almost the same performance even with noisy or small clean validation data. In Fig. 4(b)(c) we demonstrate that TopoFilter is insensitive to the parameters $k_c$ and $k_o$, up to a wide range. Here $k_c$ and $k_o$ represent the parameters for computing the $k$-nearest neighbors in largest connected component and outlier detection, respectively. This is consistent with our theoretical findings in Section 2.1. In Fig. 4(d), we show that TopoFilter is robust to the feature dimensions. See supplementary material for more results.

**Real-world corrupted dataset.** To test the effectiveness of TopoFilter in real setting, we conduct experiments on the Clothing1M dataset [47]. This dataset consists of 1 million clothing images obtained from online shopping websites with 14 classes. The labels in this dataset are extremely noisy (with an estimate accuracy of 61.54%) and their structure is unknown. This dataset also provides 50$k$, 14$k$ and 10$k$ manually verified clean data for training, validation, and testing, respectively. Following [49, 39, 44], we evaluate the classification accuracy on the 10$k$ clean data and do not use the 50$k$ clean training data; similarly, we use a randomly sampled pseudo-balanced subset as the training set, which includes about 260$k$ images. We use ResNet-50 with weights pre-trained on ImageNet and train the model with SGD. We set the batch size 32, learning rate 0.001, and preprocess the images following the same procedure in [49, 39, 44]. We train the model for 10 epochs and collect the clean data per epoch. The cost of computing $k$-nearest neighbors and connected components in data selection is about 25s.

We compare our method with the following ones: (1) *Standard*; (2) *Forward Correction* [33]; (3) *D2L* [25] (4) *Joint Optimization* [39]; (5) *PENCIL* [49]; (6) *MLNT* [23]; (7) *DY* [1]; (8) *GCE* [52]; (9) *SL* [44]. As is shown in Table 2, TopoFilter obtains the best performance compared to the baseline methods. This demonstrates its applicability to the real-world scenarios beyond the synthetic noise.

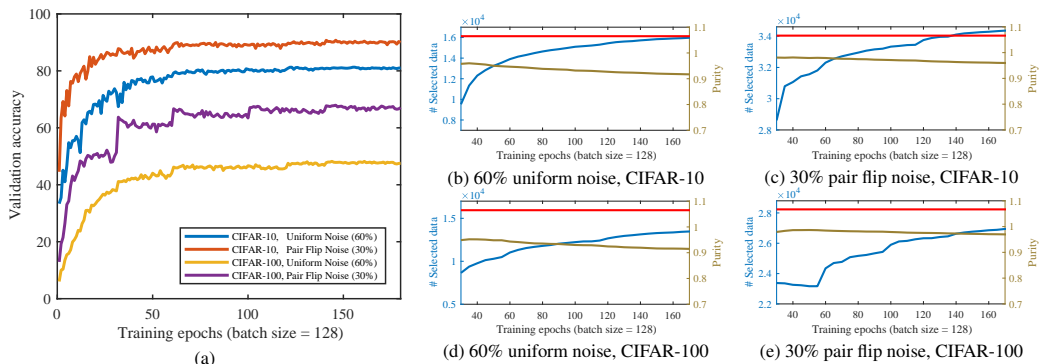

Figure 3: (a) Validation accuracies. (b-e) The size of selected data (the blue curve) and its purity (the brown curve). The red line denotes the upper-bound size of clean data.

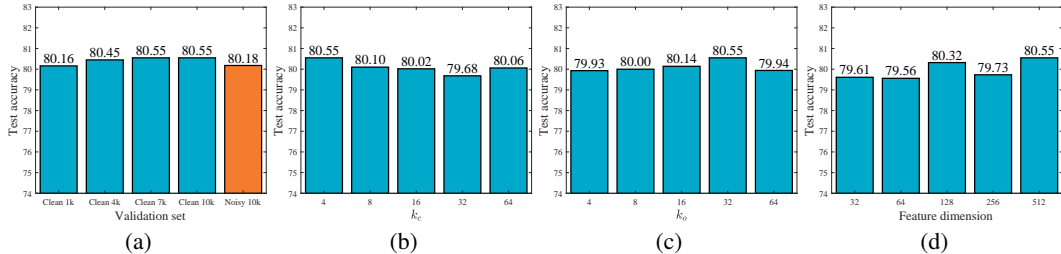

Figure 4: Parameter analysis: (a) validation set; (b) $k_c$; (c) $k_o$; (d) Feature dimension. We use uniform flipping noise (60%) and CIFAR-10. For each figure, we change one of the parameters while keeping the others fixed. (to $k_c = 4$, $k_o = 32$, feature dimension = 512, validation set = clean 10k).

## 4 Conclusion

We propose a novel method named TopoFilter for the learning with label noise. Our method leverages the topological property of the data in feature space, and jointly learns the data representation and collects the clean data during training. Theoretically, we show that TopoFilter is able to select the most of clean data with high confidence. Our empirical results on different datasets demonstrate the advantages of TopoFilter in improving the robustness of deep models to label noise.

We note that this paper only focuses on the connected components of the data in the latent representational space. In the future, we may extend the algorithm to a differentiable topological loss [9] based on the theory of persistent homology [11]. The theory has been shown to provide robust solution to learning problems such as weakly supervised learning [19], clustering [32, 8], and graph neural networks [53].

## Broader Impact

Label noise is ubiquitous in real-world data. This noise may arise from the cheap but imperfect annotations, such as crowdsourcing and online queries. Moreover, even by human annotators, the data labeling process is still error-prone. Another typical source of noise is the data poisoning, where corruptions are intentionally injected into the labels. Training with noisy labels would severely deteriorate the performance of deep models, due to their strong memorization ability and overfitting on corrupted information [51, 2]. Therefore, limiting the adverse influence of noisy labels is of great practical significance and has gained increasing attention from the community.

In this work we attack the label noise from the perspective of data topology. Different from previous works which mostly inspect the sample losses or predicted posteriors, we show that the spatial behavior of the data could be well exploited, a point that has been largely ignored before. Importantly, in theory we prove that our topology-motivated method is able to exhaustively select the clean data with high probability. In this way we keep the network away from the negative influence of corrupted labels and promote the training healthily and steadily. Our method is simple yet with theoretical insights, and would provide contributions supplementary to the existing works. We believe it deserves the attention from the machine learning community.

## Acknowledgement

Zheng and Chen's research was partially supported by NSF CCF-1855760 and IIS-1909038. Wu and Metaxas's research was partially supported by NSF CCF-1733843, IIS-1703883, CNS-1747778, IIS-1763523, IIS-1849238-825536 and MURI-Z8424104-440149. Goswami's research was partially supported by NSF CRII-1755791 and CCF-1910873.

## Footnotes

[2]Code is available at https://github.com/pxiangwu/TopoFilter

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
