[Supplementary Material]

# A Topological Filter for Learning with Label Noise
## – Supplementary Material –

In this supplemental material, we first provide proofs of all the major theorems and lemmas in the main paper. Then we discuss the choice of hyper-parameters, mainly the $\zeta$-filtering parameter $\zeta$. We also discuss other relevant hyper-parameters. Finally, we provide additional results from the data domain different than images.

## A   Proofs of Purity and Abundancy of TopoFilter

We provide proofs of all theorems and lemmas in the main paper. For completeness, we restate all the definitions, theorems and lemmas. Theorem 1 provides guarantees for the purity of the selected data. Theorem 2 shows the abundancy, i.e., our algorithm collects sufficient amount clean data.

**Background and Setting**

For the convenience of the reader, we restate our notation here. Assume that the data points and labels lie in $\mathcal{X} \times \mathcal{Y}$, where the features $\mathcal{X} \subset R^d$ and labels $\mathcal{Y} := [\mathcal{C}] := \{1, 2, 3, \cdots, \mathcal{C}\}$. Assume the (data, true label) pairs follow some distribution $\mathcal{F} \sim \mathcal{X} \times \mathcal{Y}$. Let $f(\boldsymbol{x}) := \sum_{i \in [\mathcal{C}]} \mathcal{F}(\boldsymbol{x}, i)$ be the density at $\boldsymbol{x}$. Due to label noise, label $y = i$ is flipped to $\widetilde{y} = j$ with probability $\tau_{ij}$ and is assumed to be independent of $\boldsymbol{x}$.

Let $\mathbf{X} \subset \mathcal{X}$ be the finite set of features in the data sample, and let $G(\mathbf{X}, k)$ be the mutual $k$-nearest neighbor graph on $\mathbf{X}$ using the Euclidean metric on $\mathcal{X}$, whose edge set $E = \{(\boldsymbol{x}_1, \boldsymbol{x}_2) \in \mathbf{X}^2 \mid \boldsymbol{x}_1 \in KNN(\boldsymbol{x}_2) \text{ or } \boldsymbol{x}_2 \in KNN(\boldsymbol{x}_1)\}$. Also, $\forall i \in [\mathcal{C}]$, let $G_i(\mathbf{X}, k)$ be the induced subgraph of $G(\mathbf{X}, k)$ consisting only of vertices $\boldsymbol{x} \in \mathbf{X}$ with label $\widetilde{y}(\boldsymbol{x}) = i$.

Let $\eta_i(\boldsymbol{x}) = P(y = i \mid \boldsymbol{x})$ and $\widetilde{\eta}_i(\boldsymbol{x}) = P(\widetilde{y} = i \mid \boldsymbol{x})$ be the clean and noisy posterior probability of labels given a feature $\boldsymbol{x}$, respectively. For simplicity, we focus on the binary label case for now. Then for $i \in \{0, 1\}$, these two probabilities are related by $\widetilde{\eta}_i(\boldsymbol{x}) = (1 - \tau_{01} - \tau_{10})\eta_i(\boldsymbol{x}) + \tau_{1-i,i}$. Define the super level set $L(t) = \{\boldsymbol{x} \mid \max(\eta_1(\boldsymbol{x}), \eta_0(\boldsymbol{x})) \geq t\}$. For binary case, we have a partition of the space:

$$
\begin{aligned}
A_i^+ &= \left\{ \boldsymbol{x} : \widetilde{\eta}_i(\boldsymbol{x}) > \max\left(\tfrac{1}{2}, \tfrac{1+\tau_{i,1-i}-\tau_{1-i,i}}{2}\right) \right\} = \left\{ \boldsymbol{x} : \eta_i(\boldsymbol{x}) > \max\left(\tfrac{1}{2}, \tfrac{1/2-\max(\tau_{10},\tau_{01})}{2(1-\tau_{10}-\tau_{01})}\right) \right\}, \\
A_i^- &= \left\{ \boldsymbol{x} : \widetilde{\eta}_i(\boldsymbol{x}) < \min\left(\tfrac{1}{2}, \tfrac{1+\tau_{i,1-i}-\tau_{1-i,i}}{2}\right) \right\} = \left\{ \boldsymbol{x} : \eta_i(\boldsymbol{x}) < \min\left(\tfrac{1}{2}, \tfrac{1/2-\max(\tau_{10},\tau_{01})}{2(1-\tau_{10}-\tau_{01})}\right) \right\}, \\
A^b &= \mathcal{X} \setminus (A_i^+ \cup A_i^-).
\end{aligned}
$$

We also restate the definition of purity here. Consider an algorithm $\mathcal{A}$ that takes as input a random sample of size $n$, $S_n = \{(\boldsymbol{x}_i, \widetilde{y}(\boldsymbol{x}_i))\}_{i=1}^n$, and let $\mathbf{X} := \{\boldsymbol{x}_i\}_{i=1}^n \subset \mathcal{X}$. Algorithm $\mathcal{A}$ then outputs $\cup_{i \in \{0,1\}} C^{(i)}$, where $C^{(i)} \subseteq \mathbf{X}_i := \{\boldsymbol{x} : \widetilde{y}(\boldsymbol{x}) = i\}$ is the claimed "clean" set for label $i$.

**Definition 1** (**Purity**). *We define two kinds of purity of $\mathcal{A}$ on $S_n$. One captures the worst-case behavior of the algorithm, while the other captures the average-case behavior.*

*1. **Minimum Purity*** $\ell_{S_n, \mathcal{A}} := \min_{i \in \{0,1\}} \min_{\boldsymbol{x} \in C^{(i)}} P(y = i \mid \widetilde{y} = i, \boldsymbol{x}) = \min_{i \in \{0,1\}} \min_{\boldsymbol{x} \in C^{(i)}} \tau_{ii} \frac{\eta_i(\boldsymbol{x})}{\widetilde{\eta}_i(\boldsymbol{x})}.$

**2. Average Purity** $\ell'_{S_n,\mathcal{A}} := \sum_{i \in \{0,1\}} \frac{1}{|C^{(i)}|} \sum_{\boldsymbol{x} \in C^{(i)}} \tau_{ii} \frac{\eta_i(\boldsymbol{x})}{\tilde{\eta}_i(\boldsymbol{x})}$.

**Assumption.**

- **A1:** $f(\boldsymbol{x})$ (the density on the feature space) has compact support.

- **A2:** $\forall i \in \{0,1\}$, $\eta_i(\boldsymbol{x})$ is continuous.

- **A3:** $\forall i \in \{0,1\}$, $A_i^+$ is a connected set.

- **A4:** $\tau_{10}, \tau_{01} \in \left[0, \frac{1}{2}\right)$

Denote by $\mathcal{A}_0$ the naive algorithm which takes input $S_n$ and simply outputs $C^{(i)} = \mathbf{X}_i$ for $i = 0, 1$, i.e., does no processing and treats corrupted labels as clean. The purity of $\mathcal{A}_0$ is the "default" purity of the data set. Denote our algorithm with parameter $\zeta$ by $\mathcal{A}_\zeta$. Let $\zeta' = \frac{1}{2}\left(\zeta + \frac{1+|\tau_{10}-\tau_{01}|}{2}\right)$, and $e$ be the natural constant.

**Theorem 1 (Purity Guarantee).** $\forall \delta > 0$, $\forall \zeta > \frac{1+|\tau_{10}-\tau_{01}|}{2}$ and $\forall q > 1$, there exist $N(\delta, \zeta, q) > 0$, $c_1(\zeta) > 0$, constant $c_2 \in \left(0, \frac{e-1}{e}\right)$, and an increasing function $g_1(\zeta) \in \left[\frac{[2\zeta+1+|\tau_{10}-\tau_{01}|-4\max(\tau_{10},\tau_{01})]\min(\tau_{11},\tau_{00})}{[2\zeta+1+|\tau_{10}-\tau_{01}|](1-\tau_{10}-\tau_{01})}, 1\right]$ and function $g_2(\zeta) > 0$, such that $\forall n \geq N$ and $\forall k \in [c_1(\zeta)\log^q n, c_2 n]$:

*1.* $P\left[(\ell_{S_n,\mathcal{A}_\zeta} - \ell_{S_n,\mathcal{A}_0}) > g_1(\zeta)\right] \geq 1 - \delta$, *and*

*2.* $P\left[(\ell'_{S_n,\mathcal{A}_\zeta} - \ell'_{S_n,\mathcal{A}_0}) > g_2(\zeta)\right] \geq 1 - \delta$.

To proceed, we first state a lemma from [1] that we use. This lemma shows that certain lower (upper) bounds on the true density in a ball imply lower (upper) bounds on the empirical density of a ball. We mention that we will also use certain proof techniques from [2] that are help to analyze clustering using KNN.

**Lemma 0** (Lemma 7 in (Kamalika et al., 2010)). *Assume* $k \geq d\log n$ *and fix some* $\delta > 0$. *Then there exists a constant* $c_0$ *such that with probability* $1 - \delta$, *every ball* $B \subset R^d$ *satisfies the following conditions:*

$$P(B) \geq \frac{2C_0 \log(2/\delta)\log n}{n} \qquad \implies \qquad P_n(B) > 0$$

$$P(B) \geq \frac{k}{n} + \frac{2C_0 \log 2/\delta}{n}\sqrt{kd\log n} \qquad \implies \qquad P_n(B) > \frac{k}{n}$$

$$P(B) \leq \frac{k}{n} - \frac{2C_0 \log 2/\delta}{n}\sqrt{kd\log n} \qquad \implies \qquad P_n(B) < \frac{k}{n}$$

*Here* $f_n(B) = \frac{|X_n \cap B|}{n}$ *is the empirical mass of* $B$, *while* $f(B) = \int_{\boldsymbol{x} \in B} f(\boldsymbol{x})d\boldsymbol{x}$ *is its true mass.*

For more detail about this Lemma, please refer to [1]. Using Lemma 0, we can show that by picking certain $k$ and $n$, all data point from region $L(\zeta)$ connected in the symmetric KNN graph.

Define for $i \in \{0,1\}$, $\mathbf{X}_i(t) = L(t) \cap \mathbf{X}_i$.

**Lemma 1 (Connectivity).** $\forall \delta > 0$, $\forall t \in [0, 1)$, *there exist constants* $N(\delta, t) > 0$, *and* $c_1(t) > 0$ *such that* $\forall n \geq N(\delta, t)$, $\forall i \in \{0,1\}$, $\forall q > 1$, *and* $\forall k > c_1(t)\log^q(n)$, $\mathbf{X}_i(t)$ *is connected in* $G_i(\mathbf{X}, k)$ *with probability at least* $1 - \delta$.

*Proof.* We first develop some notation. Let $V_d$ be the volume of the unit d-dimensional ball. Let $\mu_s(r) = V_d r^d \min_{i \in \{0,1\}} \min_{\boldsymbol{x} \in L(t) \cap A_i^+} [p_{ii}(\boldsymbol{x}) + p_{1-i,i}(\boldsymbol{x})]$ and $\mu_l(r) = V_d(2r)^d \max_{i \in \{0,1\}} \max_{x \in L(t) \cap A_i^+} f(x)$.

Fix any $\delta > 0$. We will prove the lemma by showing that there exist $C_0 > 0$, $N(\delta, t) > 0$ and $r \in \left(0, \left(\frac{\log^q n}{n}\right)^{1/d}\right]$, $q > 1$ such that $\forall n \geq N(\delta, t)$

$$\mu_s(r) \geq \frac{2C_0 \log 4/\delta \log n}{n}, \text{ and}$$

$$\mu_l(r) \leq \frac{k}{n} - \frac{2C_0 \log 4/\delta}{n}\sqrt{kd \log n}, \text{ and}$$

$$k > \max\left(d \log n, \ 4dC_0^2 \log^2\left(4/\delta\right) \log n + \frac{2\mu_l(r)}{r^d}\right).$$

As a consequence, we will conclude that with probability at least $1 - \delta$, we have $X_i(t)$ is connected in $G_i(\mathbf{X}, k)$.

Since $f(\boldsymbol{x})$ has compact support, $L(t) = \left\{\boldsymbol{x} \mid \max_{i \in \{0,1\}} \eta_i(\boldsymbol{x}) \geq t\right\}$ is a closed subset of the domain. Then $L(t)$ is compact. For $\forall r \in \left(0, \left(\frac{\log^q n}{n}\right)^{1/d}\right]$, we have $L(t) \subset \bigcup_{j=1}^{m} B_j(r)$. From now, we fix some $r \in \left(0, \left(\frac{\log^q n}{n}\right)^{1/d}\right]$.

For a data point $\boldsymbol{x}$, its KNN radius is the distance to its kth nearest neighbor. Define $R^*$ to be the minimum KNN radius for any $\boldsymbol{x} \in X_i(t)$, and further define two events $E_1$ and $E_2$ as:

$$E_1 = \{\exists \boldsymbol{x} \in \mathbf{X} \cap B_j(r), \widetilde{y} = i, \forall j \in [m]\}$$

$$E_2 = \{R^* > 2r\}$$

Then for the statement $E = \{X_i(t) \text{ is connected}\}$ we have $E_1 \cap E_2 \subset E$ and thus $f(E) \geq f(E_1 \cap E_2) = 1 - f(E_1^c \cup E_2^c) \geq 1 - f(E_1^c) - f(E_2^c)$. This is true because for every $B(r)$ we will see at least one type $i$ point. But for every $B(2r)$ we will have fewer than $k$ points, which implies that all these points will be the nearest neighbor of each other in $B(2r)$. For a $2r$ diameter of $B(2r)$, we can juxtapose two $B(r)$ with the diameter pass both of their center. Within each of these two $B(r)$, we will see at least one type-$i$ point(See Fig 1). Thus $E_1 \cap E_2$ implies $\bigcup_{j=1}^{m} B_j(r)$ is connected , which then implies $E$.

Figure 1: Demonstration for $E_1 \cap E_2 \rightarrow E$. Green points are type i points. Every points in the large ball are KNN to each other, since the maximum KNN radius is larger than $2r$.

Suppose there are d-dimensional balls $B_s$ and $B_l$ whose measure are $\mu_s$ and $\mu_l$ separately. We can pick proper $n$ and $k$ such that conditions in Lemma 0 will be satisfied :

$$P(B_s) \geq \frac{2C_0 \log 4/\delta \log n}{n} \qquad \Longrightarrow \qquad P_n(B_s) > 0$$

$$P(B_l) \leq \frac{k}{n} - \frac{2C_0 \log 4/\delta}{n}\sqrt{kd \log n} \qquad \Longrightarrow \qquad P_n(B_l) < \frac{k}{n}$$

To see this, we could first pick large $n$, such that the first inequality is fulfilled. Then we increase $k$ (the RHS of inequality 2 is increasing with respect to k for fixed n and for $k > \frac{4C_0^2 \log^2\left(4/\delta\right)d \log n}{2}$) such that the second inequality is fulfilled. The desired k should be:

$$k > \left[ \frac{2C_0 \log\left(4/\delta\right)\sqrt{d \log n} + \sqrt{4C_0^2 \log^2\left(4/\delta\right)d\log n + 2n\mu_l}}{2} \right]^2$$

We observe that $k > 4dC_0^2 \log\left(4/\delta\right)^2 \log n + 2n\mu_l(r)$ satisfies the above inequality.

We note that $2n\mu_l(r) = 2nV_d r^d \max_{i\in\{0,1\}} \max_{\boldsymbol{x}\in L(t)\cap A_i^+} f(\boldsymbol{x})$, and because $r < \left(\frac{\log^q n}{n}\right)^{1/d}$, this is smaller than $2V_d \max_{i\in\{0,1\}} \max_{\boldsymbol{x}\in L(t)\cap A_i^+} f(\boldsymbol{x}) \log^q n$. As a result, if we set $k > 4dC_0^2 \log^2\left(4/\delta\right)\log n + 2V_d \max_{i\in\{0,1\}} \max_{\boldsymbol{x}\in L(t)\cap A_i^+} f(\boldsymbol{x}) \log^q n$, $\forall q > 1$, then this value of $k$ satisfies the inequality for all $r$.

As a result, if we take $k > \max\left( d\log n, 4dC_0^2 \log^2\left(4/\delta\right)\log n + 2^{d+1}V_d \max_{\boldsymbol{x}\in L(t)\cap A_i^+} f(\boldsymbol{x}) \log^q n \right)$ $\forall q > 1$, then replacing $\delta$ by $\delta/2$ in Lemma 0, $P[E_1]$ and $P[E_2]$ are both at least $1 - \delta/2$. Thus $f(E) > 1 - f(E_1^c) - f(E_2^c) > 1 - P\left[P_n(B_s) \leq 0\right] - P\left[P_n(B_l) \geq \frac{k}{n}\right] \geq 1 - \delta$, completing the proof. $\qquad\square$

We also restate notations that needed by Lemma 2 here. Remember that $\zeta' = \frac{1}{2}\left(\zeta + \frac{1+|\tau_{10}-\tau_{01}|}{2}\right)$. Define $\mathbf{X}_i^c(\zeta') := \overline{L(\zeta')^c}\cap\mathbf{X}_i$. Define $r_0^{(i)} = \min\left\|\boldsymbol{x}_1^{(i)} - \boldsymbol{x}_2^{(i)}\right\|$ for $\boldsymbol{x}_1^{(i)} \in X_i(\zeta)$ and $\boldsymbol{x}_2^{(i)} \in X_i^c(\zeta')$. Let $V_d$ to be the volume of $d$-dimensional unit ball. Let $p_\zeta^{(i)} := \min_{\boldsymbol{x}\in L(\zeta)\cap A_i^+} f(\boldsymbol{x})V_d(r_0^{(i)})^d$ and $p_{\zeta'}^{(i)} := \min_{\boldsymbol{x}\in L(\zeta')^c\cap A_i^{+c}} f(\boldsymbol{x})V_d(r_0^{(i)})^d$. Since $f(\boldsymbol{x})$ has compact support, $A_i^{+c}$ is closed and $L(\zeta')^c \subset A_i^{+c}$, then $p_\zeta^{(i)} > 0$ and $p_{\zeta'}^{(i)} > 0$.

Let $K(p \parallel q)$ be the KL divergence between distribution $p$ and $q$.

**Lemma 2 (Isolation).** $\forall\delta > 0$, $\forall\zeta > \frac{1+|\tau_{10}-\tau_{01}|}{2}$, *there exists constant* $c_2 \in \left(0, \frac{e-1}{e}\right)$, $N(\delta,\zeta) > 0$ *such that* $\forall n \geq N(\delta,\zeta)$, $\forall k < c_2(\zeta)\left[\min_{i\in\{0,1\}} \min\left(p_\zeta^{(i)}, p_{\zeta'}^{(i)}\right)(n-1)\right] + 1$ *and* $\forall i \in \{0,1\}$:

$$P\left(\nexists edge = (u,v) \in G_i(\mathbf{X}, k) : u \in \mathbf{X}_i(\zeta), v \in \mathbf{X}_i^c(\zeta')\right) \geq 1 - \delta.$$

*Proof.* Let $E$ be the event $\{\nexists edge = (u,v) \in G_i(\mathbf{X}, k) : u \in \mathbf{X}_i(\zeta), v \in \mathbf{X}_i^c(\zeta')\}$. Let $R(\boldsymbol{x})$ be the nearest neighbor radius of point $\boldsymbol{x}$, which is the distance from $\boldsymbol{x}$ to its $k$th nearest neighbor. Then let $R_\zeta^* = \max_{i\in\{0,1\}} \max_{\boldsymbol{x}\in\mathbf{X}_i(\zeta)} R(\boldsymbol{x})$ and $R_{\zeta'}^* = \max_{i\in\{0,1\}} \max_{\boldsymbol{x}\in\mathbf{X}_i^c(\zeta')} R(\boldsymbol{x})$ separately. Let $r_0 = \min_{i\in\{0,1\}} r_0^{(i)}$. Let $p_\zeta = \min_{i\in\{0,1\}} p_\zeta^{(i)}$ and $p_{\zeta'} = \min_{i\in\{0,1\}} p_{\zeta'}^{(i)}$. Let $M_\zeta \sim Bin(n-1, p_\zeta)$ and $M_{\zeta'} \sim Bin(n-1, p_{\zeta'})$. Then:

$$P(E) \geq P\left(\{R_\zeta^* \leq r_0\} \cap \{\{R_{\zeta'}^* \leq r_0\}\right) \geq 1 - P(R_\zeta^* > r_0) - P(R_{\zeta'}^* > r_0) \tag{1}$$

$$\geq 1 - P\left(\bigcup_{\boldsymbol{x} \in \mathbf{X}_i(\zeta)} \{R(\boldsymbol{x}) > r_0\}\right) - P\left(\bigcup_{\boldsymbol{x} \in \mathbf{X}_i^c(\zeta')} \{R(\boldsymbol{x}) > r_0\}\right) \tag{2}$$

$$\geq 1 - n_\zeta \mu[L(\zeta)] P(M_\zeta \leq k-1) - n_{\zeta'} \mu[L(\zeta')^c] P(M_{\zeta'} \leq k-1) \tag{3}$$

$$\geq 1 - n_\zeta \mu[L(\zeta)] \exp\left\{-(n-1)K\left(\frac{k-1}{n-1} \,\middle\|\, p_\zeta\right)\right\} - n_{\zeta'} \mu[L(\zeta')^c] \exp\left\{-(n-1)K\left(\frac{k-1}{n-1} \,\middle\|\, p_{\zeta'}\right)\right\} \tag{4}$$

$$\geq 1 - n_\zeta \mu[L(\zeta)] \exp\left\{-(n-1)\left[\frac{(e-1)p_\zeta}{e} - \frac{k-1}{n-1}\right]\right\} - n_{\zeta'} \mu[L(\zeta')^c] \exp\left\{-(n-1)\left[\frac{(e-1)p_{\zeta'}}{e} - \frac{k-1}{n-1}\right]\right\} \tag{5}$$

$$\geq 1 - 2\max(n_\zeta \mu[L(\zeta)], n_{\zeta'} \mu[L(\zeta')^c]) \exp\left\{-(n-1)\left[\frac{(e-1)\min(p_\zeta, p_{\zeta'})}{e} - \frac{k-1}{n-1}\right]\right\} \tag{6}$$

For inequality (3), we use Chernoff lower tail inequality again, which require $\frac{k-1}{n-1} < \min(p_\zeta, p_{\zeta'})$. Inequality (4) holds because $K(\frac{k-1}{n-1} \| p_\zeta) = \frac{k-1}{n-1}\ln\left(\frac{k-1}{p_\zeta(n-1)}\right) + \frac{n-k}{n-1}\ln\left(\frac{n-k}{(1-p_\zeta)(n-1)}\right) \geq \frac{k-1}{n-1}\ln\left(\frac{k-1}{p_\zeta(n-1)}\right) + p_\zeta - \frac{k-1}{n-1} \geq -\frac{p_\zeta}{e} + p_\zeta - \frac{k-1}{n-1} \geq \frac{(e-1)p_\zeta}{e} - \frac{k-1}{n-1}$. This is also true for $K(\frac{k-1}{n-1} \| p_{\zeta'})$. The first inequality comes from the fact that $\ln(\boldsymbol{x}) \geq (\boldsymbol{x}-1)/\boldsymbol{x}$. And the second inequality comes from the fact that $-\frac{p_\zeta}{e}$ is the minimizer of $\frac{k-1}{n-1}\ln\left(\frac{k-1}{p_\zeta(n-1)}\right)$ with respect to $\frac{k-1}{n-1}$.

Now let $c_2 < (e-1)/e$, and $k \leq c_2\left[\min_{i\in\{0,1\}} \min\left(p_\zeta^{(i)}, p_{\zeta'}^{(i)}\right)(n-1)\right] + 1$ as in the statement of the lemma. Then we have that $\frac{(e-1)\min(p_\zeta, p_{\zeta'})}{e} - \frac{k-1}{n-1} \geq \left(\frac{e-1}{e} - c_2\right)\min(p_\zeta, p_{\zeta'})$ is independent of $n$. Then as $n \to \infty$, $\exp\left\{-(n-1)\left[\frac{(e-1)\min(p_\zeta, p_{\zeta'})}{e} - \frac{k-1}{n-1}\right]\right\} \to 0$, and thus $P[E] \to 1$.

$\square$

Lemma 2 tells us, if $k$ is properly set, $X_i(\zeta)$ and $X_i^c(\zeta')$ are separated. So the remaining cases for points in region $L(\zeta')^c$ are case where all its neighbors are located in $L(\zeta')^c$ and case where part of its neighbors are in $L(\zeta')\backslash L(\zeta)$. Next lemma will show, if we filter out points that don't have enough desired number of neighbors, we will guarantee there are no points in $L(\zeta')^c$ remaining in $\bigcup_{i\in\{0,1\}} C^{(i)}$.

**Lemma 3** ($\zeta$-filtering). $\forall \delta > 0$ and $\zeta \in \left(\frac{1+|\tau_{10}-\tau_{01}|}{2}, 1\right)$, there exists $N(\delta, \zeta) > 0$ and $c_3(\zeta) > 0$, such that $\forall n \geq N$, $k > c_3(\zeta)\log(2n/\delta)$ and $\forall i \in \{0, 1\}$ then:

$$P\left(C^{(i)}(\zeta) \cap L(\zeta')^c = \emptyset\right) \geq 1 - \delta.$$

*Proof.* Let $\{\boldsymbol{x}^{(z)}\}_{z=1}^k$ be the set of k nearest neighbors of $\boldsymbol{x}$, and consider an $\boldsymbol{x}$ such that for all $1 \leq z \leq k$, $\boldsymbol{x}^{(z)} \in G_i(\mathbf{X}, k) \cap L(\zeta')^c$. Let $N^{(i)}(\boldsymbol{x})$ be the number of type 1 ($\widetilde{y} = 1$) nearest neighbors of such an $\boldsymbol{x}$. We know $N^{(i)}(\boldsymbol{x}) = \sum_{z=1}^k \text{Bernoulli}(\widetilde{\eta}(\boldsymbol{x}^{(z)}))$. Since $\widetilde{\eta}(\boldsymbol{x}^{(z)}) \leq p^* := \zeta'$ for all $1 \leq z \leq k$, we observe that $N^{(i)}(\boldsymbol{x})$ is stochastically dominated by $M := \text{Binomial}(k, p^*)$.

By Lemma 2, $\forall \delta > 0$, $\forall \boldsymbol{x} \in L(\zeta')^c$, for all $1 \leq z \leq k$, $\boldsymbol{x}^{(z)} \notin L(\zeta)$ with probability at least $1 - \delta/2$. For convenience, we denote the event that Lemma 2 holds as $E_I$. Therefore, with probability at least $1 - \delta/2$, $N^{(i)}(\boldsymbol{x})$ is well-defined $\forall \boldsymbol{x} \in L(\zeta')^c$.

$$P\left(C^{(i)}(\zeta) \cap L(\zeta')^c = \emptyset \mid E_I\right) = 1 - P\left(C^{(i)} \cap L(\zeta')^c \neq \emptyset \mid E_I\right) \tag{7}$$

$$= 1 - P\left(\bigcup_{\boldsymbol{x} \in C^{(i)} \cap L(\zeta')^c} \left\{N^{(i)}(\boldsymbol{x}) > \lceil \zeta k \rceil\right\}\right), \tag{8}$$

where $\zeta$ is the threshold used in the algorithm to filter outliers in the largest connected component. Let $n_{\zeta'}^{(i)} := \#\mathbf{X}_i^c(\zeta')$. Continuing, we have

$$1 - P\left(\bigcup_{\boldsymbol{x} \in C^{(i)}(\zeta') \cap L(\zeta')^c} \left\{N^{(i)}(\boldsymbol{x}) > \lceil \zeta k \rceil\right\}\right) \geq 1 - n_{\zeta'}^{(i)} P\left(N^{(i)}(\boldsymbol{x}) > \lceil \zeta k \rceil\right) \tag{9}$$

$$\geq 1 - n_{\zeta'}^{(i)} P\left(M > \lceil \zeta k \rceil\right) \tag{10}$$

$$\geq 1 - n_{\zeta'}^{(i)} \exp\left\{-kK(\zeta \parallel \zeta')\right\} \tag{11}$$

inequality 11 uses the Chernoff tail bound. Since $\zeta' = \frac{1}{2}\left(\zeta + \frac{1+|\tau_{10}-\tau_{01}|}{2}\right) < \zeta$ for all $\zeta > \frac{1+|\tau_{10}-\tau_{01}|}{2}$. Define $c_4 = K(\zeta \parallel \zeta')$. After choosing a large enough $n_{\zeta'}^{(i)}$, given any $\delta > 0$, we set $k > \frac{1}{c_4} \log\left(2n_{\zeta'}^{(i)}/\delta\right)$, and we have that $1 - n_{\zeta'}^{(i)} \exp\left\{-kK(\zeta \parallel \zeta')\right\} > 1 - \delta/2$. Denote the event $P\left(C^{(i)} \cap L(\zeta')^c = \emptyset\right)$ as $E_F$; we have that,

$$P\left(C^{(i)} \cap L(\zeta')^c = \emptyset\right) = P(E_F \cap E_I) = 1 - P(E_I^c) - P(E_F^2 \mid E_I) = 1 - \delta/2 - \delta/2 = 1 - \delta$$

$$\square$$

Now we are ready to prove Theorem 1. Lemma 1 tells us that as long as we take moderate $k$, all points in $L(\zeta)$ will be connected in the induced sub-graph. And Lemma 2 and Lemma 3 say that after removing points whose degree doesn't coincide its label, we will have no points from $L(\zeta')^c$ remained in $\bigcup_i C^{(i)}$. Then we use the value of the $\widetilde{\eta}_i(\boldsymbol{x})$ at region $L(\zeta)$ to prove our main theorem.

### A.1 Proof of the Purity Theorem

*Proof.* By combining Lemmas 1, 2 and 3, we have that $\forall \delta > 0$ and $\forall \zeta \in \left(\frac{1+|\tau_{10}-\tau_{01}|}{2}, 1\right)$, $\exists N(\delta, \zeta) > 0$, such that $\forall n \geq N(\delta)$, each of the following event holds with probability at least $1 - \delta/4$:

$$\begin{array}{lll} \text{Connectivity} & E_C & = & \{\mathbf{X} \cap L(\zeta) \text{ is connected}\} \\ \text{Isolation} & E_I & = & \{\nexists edge = (u, v) \in G_i(\mathbf{X}, k) : u \in \mathbf{X}_i(\zeta), v \in \mathbf{X}_i^c(\zeta'), \forall i \in \{0, 1\}\} \\ \text{Filtering} & E_F & = & \{\bigcup_{i \in \{0, 1\}} C^{(i)} \cap L(\zeta')^c = \emptyset\} \end{array}$$

First we prove the theorem for the minimum purity $\ell_{S_n, \mathcal{A}}$, assuming all of the above events. For the minimum purity, we will assume that $\forall i \in \{0, 1\}$, $\min_{\boldsymbol{x} \in \mathcal{X}} \eta_i(\boldsymbol{x}) = 0$[1]. In the following, $\xrightarrow{p}$ will

denote convergence in probability, and $\xrightarrow{f}$ will denote convergence in distribution.

$$\ell_{S_n,\mathcal{A}_\zeta} = \min_{i\in\{0,1\}} \min_{\boldsymbol{x}\in C^{(i)}\cap L(\zeta')} \tau_{ii}\frac{\eta_i(\boldsymbol{x})}{\widetilde{\eta}_i(\boldsymbol{x})} \xrightarrow{p} \min_{i\in\{0,1\}} \min_{\boldsymbol{x}\in L(\zeta')} \tau_{ii}\frac{\eta_i(\boldsymbol{x})}{\widetilde{\eta}_i(\boldsymbol{x})} \tag{12}$$

$$= \min_{i\in\{0,1\}} \min_{\boldsymbol{x}\in L(\zeta')} \tau_{ii}\frac{\widetilde{\eta}_i(\boldsymbol{x})-\tau_{1-i,i}}{(1-\tau_{10}-\tau_{01})\widetilde{\eta}_i(\boldsymbol{x})} = \frac{[\zeta'-\max(\tau_{10},\tau_{01})][\min(\tau_{11},\tau_{00})]}{\zeta'(1-\tau_{10}-\tau_{01})} \tag{13}$$

$$\ell_{S_n,\mathcal{A}_0} = \min_{i\in\{0,1\}} \min_{\boldsymbol{x}\in C^{(i)}\cap\mathcal{X}} \tau_{ii}\frac{\eta_i(\boldsymbol{x})}{\widetilde{\eta}_i(\boldsymbol{x})} \xrightarrow{p} \min_{i\in\{0,1\}} \min_{\boldsymbol{x}\in\mathcal{X}} \tau_{ii}\frac{\eta_i(\boldsymbol{x})}{\widetilde{\eta}_i(\boldsymbol{x})} \tag{14}$$

$$= \min_{i\in\{0,1\}} \min_{\boldsymbol{x}\in A^-} \tau_{ii}\frac{\eta_i(\boldsymbol{x})}{\widetilde{\eta}_i(\boldsymbol{x})} = 0 \tag{15}$$

To show the convergence in probability in (12), denote $g_i(\boldsymbol{x}) = \tau_{ii}\frac{\eta_i(\boldsymbol{x})}{\widetilde{\eta}_i(\boldsymbol{x})}$. Let $\mathcal{F}_i$ be the cumulative distribution function of the scalar random variable $g_i(\boldsymbol{x})$. Also, let $g_{(1,n,i)} = \min_{\boldsymbol{x}\in C^{(i)}\cap L(\zeta')} g_i(\boldsymbol{x})$.

Then using the property of minimum order statistics, for all $g$ in the range of $g(\boldsymbol{x})$ where $\boldsymbol{x}\in L(\zeta')$, the cdf of $g_{(1,n,i)}$:

$$\mathcal{F}_{(1,n,i)}(g) := P\left[g_{(1,n,i)} < g\right] = 1 - P[g_{(1,n,i)} \geq g] = 1 - [P(g_i(\boldsymbol{x}) \geq g)^n] \tag{16}$$

$$= 1 - [1 - \mathcal{F}_i(g)]^n \tag{17}$$

Let $g_i^* = \min_{\boldsymbol{x}\in L(\zeta')} g_i(\boldsymbol{x})$, so that we have $\mathcal{F}_i(g^*) = 0$. We have that $\lim_{n\to\infty} \mathcal{F}_{(1,n,i)}(g) = \mathbb{1}_{\{g\geq g_i^*\}}(g)$, where $\mathbb{1}_A(\boldsymbol{x})$ is the indicator function such that $\mathbb{1}_A(\boldsymbol{x}) = 1$ if $\boldsymbol{x}\in A$ and 0 otherwise. Thus by definition $g_{(1,n,i)} \xrightarrow{f} g_i^*$. We will now use the fact that if $X_n \xrightarrow{f} c$ where $c$ is some constant, then $X_n \xrightarrow{p} c$, i.e, convergence in distribution to a constant implies convergence in probability. Then we have $g_{(1,n,i)} \xrightarrow{p} g_i^*$; in other words, $\min_{\boldsymbol{x}\in C^{(i)}\cap L(\zeta')} \tau_{ii}\frac{\eta_i(\boldsymbol{x})}{\widetilde{\eta}_i(\boldsymbol{x})} \xrightarrow{p} \min_{\boldsymbol{x}\in L(\zeta')} \tau_{ii}\frac{\eta_i(\boldsymbol{x})}{\widetilde{\eta}_i(\boldsymbol{x})}$.

Similarly we can show the convergence in probability in (14). Finally we plug in $\min_{\boldsymbol{x}\in A^-} \eta_i(\boldsymbol{x}) = \min_{\boldsymbol{x}\in\mathcal{X}} \eta_i(\boldsymbol{x}) = 0$.

Now we analyze the probability that the minimum purity guarantee assertion holds. Let $E_P = \left\{\ell_{S_n,\mathcal{A}_\zeta} - \ell_{S_n,\mathcal{A}_0} > \frac{[\zeta'-\max(\tau_{10},\tau_{01})]\min(\tau_{11},\tau_{00})}{\zeta'(1-\tau_{10}-\tau_{01})} \mid E_C, E_I, E_F\right\}$. By the above convergence in probability, $\forall\delta > 0$ and $\forall\zeta > \frac{1+|\tau_{10}-\tau_{01}|}{2}$, $\exists N > 0$ such that $\forall n \geq N$, $P(E_P) \geq 1 - \delta/4$. Then:

$$P(E_P \cap E_I \cap E_F \cap E_C) = P(\{E_P\} \cap \{E_F|E_I, E_C\} \cap \{E_I \mid E_C\} \cap \{E_C\})$$
$$\geq 1 - P(\{E_P^c)\}) - P(\{E_F^c \mid E_I\}) - P(\{E_I^c \mid E_C\}) - P(\{E_C^c\})$$
$$\geq 1 - 4*(\delta/4) = 1 - \delta,$$

which means that our minimum purity guarantee holds with probability at least $1 - \delta$.

Now we consider average purity (second assertion in Theorem 1). Let $h_i(\boldsymbol{x}) = \frac{\eta_i(\boldsymbol{x})}{\widetilde{\eta}_i(\boldsymbol{x})} = \frac{\widetilde{\eta}_i(\boldsymbol{x})-\tau_{1-i,i}}{(1-\tau_{10}-\tau_{01})\widetilde{\eta}_i(\boldsymbol{x})}$. Observe that $h_i(x)$ is an increasing function with respect to $\widetilde{\eta}_i(\boldsymbol{x})$. For the

average purity $\ell'_{Sn,\mathcal{A}}$ we have:

$$\ell'_{S_n,\mathcal{A}_\zeta} - \ell'_{S_n,\mathcal{A}_0} = \frac{[\zeta' - \max(\tau_{10}, \tau_{01})]\min(\tau_{11}, \tau_{00})}{\zeta'(1 - \tau_{10} - \tau_{01})} \tag{18}$$

$$= \sum_{i \in \{0,1\}} \frac{1}{|C^{(i)}|} \sum_{\boldsymbol{x} \in C^{(i)} \cap L(\zeta')} \tau_{ii} \frac{\eta_i(\boldsymbol{x})}{\widetilde{\eta}_i(\boldsymbol{x})} - \sum_{i \in \{0,1\}} \frac{1}{|C^{(i)}|} \sum_{\boldsymbol{x} \in C^{(i)} \cap \mathcal{X}} \tau_{ii} \frac{\eta_i(\boldsymbol{x})}{\widetilde{\eta}_i(\boldsymbol{x})} \tag{19}$$

$$= \sum_{i \in \{0,1\}} \frac{\tau_{ii}}{|C^{(i)}|} \left[ \sum_{\boldsymbol{x} \in C^{(i)} \cap L(\zeta')} \frac{\eta_i(\boldsymbol{x})}{\widetilde{\eta}_i(\boldsymbol{x})} - \sum_{\boldsymbol{x} \in C^{(i)} \cap \mathcal{X}} \frac{\eta_i(\boldsymbol{x})}{\widetilde{\eta}_i(\boldsymbol{x})} \right] \tag{20}$$

$$= \sum_{i \in \{0,1\}} \tau_{ii} \left[ \frac{\sum_{\boldsymbol{x} \in C^{(i)} \cap L(\zeta')} h_i(\boldsymbol{x})}{|C^{(i)}|} - \frac{\sum_{\boldsymbol{x} \in C^{(i)} \cap \mathcal{X}} h_i(\boldsymbol{x})}{|C^{(i)}|} \right] \tag{21}$$

Note that finite moment of $h_i(\boldsymbol{x})$ comes from the fact that $\widetilde{\eta}_i(\boldsymbol{x}) > \tau_{1-i,i}$, which implies that $h_i(\boldsymbol{x}) = \frac{\eta_i(\boldsymbol{x})}{\widetilde{\eta}_i(\boldsymbol{x})} < \frac{1}{\tau_{1-i,i}}$. Together with the fact that $\boldsymbol{x}$ has compact support we could show $E[h_i(\boldsymbol{x})] < \infty$. Using the law of large numbers we have:

$$\lim_{n \to \infty} (\ell'_{S_n,\mathcal{A}_\zeta} - \ell'_{S_n,\mathcal{A}_0}) = \sum_{i \in \{0,1\}} \tau_{ii} \left[ E\left[ h_i(\boldsymbol{x}) \mid \boldsymbol{x} \in L(\zeta') \right] - E\left[ h_i(\boldsymbol{x}) \right] \right] \tag{22}$$

$$= \sum_{i \in \{0,1\}} \tau_{ii} \left[ \frac{\int_{\boldsymbol{x} \in L(\zeta')} h_i(\boldsymbol{x}) f(\boldsymbol{x}) d\boldsymbol{x}}{\int_{\boldsymbol{x} \in L(\zeta')} f(\boldsymbol{x}) d\boldsymbol{x}} - \frac{\int_{\boldsymbol{x} \in \mathcal{X}} h_i(\boldsymbol{x}) f(\boldsymbol{x}) d\boldsymbol{x}}{\int_{\boldsymbol{x} \in \mathcal{X}} f(\boldsymbol{x}) d\boldsymbol{x}} \right] \tag{23}$$

$$= \sum_{i \in \{0,1\}} \tau_{ii} \left[ \frac{\int_{\boldsymbol{x} \in L(\zeta')} h_i(\boldsymbol{x}) f(\boldsymbol{x}) d\boldsymbol{x}}{\mu[L(\zeta')]} - \int_{\boldsymbol{x} \in \mathcal{X}} h_i(\boldsymbol{x}) f(\boldsymbol{x}) d\boldsymbol{x} \right] \tag{24}$$

$$= \sum_{i \in \{0,1\}} \frac{\tau_{ii}}{\mu[L(\zeta')]} \left[ [\mu[L(\zeta')] + \mu[L(\zeta')^c]] \int_{\boldsymbol{x} \in L(\zeta')} h_i(\boldsymbol{x}) f(\boldsymbol{x}) d\boldsymbol{x} - \mu[L(\zeta')] \int_{\boldsymbol{x} \in \mathcal{X}} h_i(\boldsymbol{x}) f(\boldsymbol{x}) d\boldsymbol{x} \right] \tag{25}$$

$$= \sum_{i \in \{0,1\}} \frac{\tau_{ii}}{\mu[L(\zeta')]} \left[ \mu[L(\zeta')^c] \int_{\boldsymbol{x} \in L(\zeta')} h_i(\boldsymbol{x}) f(\boldsymbol{x}) d\boldsymbol{x} - \mu[L(\zeta')] \int_{\boldsymbol{x} \in L(\zeta')^c} h_i(\boldsymbol{x}) f(\boldsymbol{x}) d\boldsymbol{x} \right] \tag{26}$$

$$= \sum_{i \in \{0,1\}} \frac{\tau_{ii}}{\mu[L(\zeta')]} \int_{\boldsymbol{x} \in L(\zeta')^c} h_i(\boldsymbol{x}) f(\boldsymbol{x}) d\boldsymbol{x} \left[ \mu[L(\zeta')^c] \frac{\int_{\boldsymbol{x} \in L(\zeta')} h_i(\boldsymbol{x}) f(\boldsymbol{x}) d\boldsymbol{x}}{\int_{\boldsymbol{x} \in L(\zeta')^c} h_i(\boldsymbol{x}) f(\boldsymbol{x}) d\boldsymbol{x}} - \mu[L(\zeta')] \right] \tag{27}$$

$$\geq \sum_{i \in \{0,1\}} \frac{\tau_{ii}}{\mu[L(\zeta')]} \int_{\boldsymbol{x} \in L(\zeta')^c} h_i(\boldsymbol{x}) f(\boldsymbol{x}) d\boldsymbol{x} \left[ \mu[L(\zeta')^c] \frac{\int_{\boldsymbol{x} \in L(\zeta')} h_i(\boldsymbol{x}) f(\boldsymbol{x}) d\boldsymbol{x}}{h_i(\zeta') \int_{\boldsymbol{x} \in L(\zeta')^c} f(\boldsymbol{x}) d\boldsymbol{x}} - \mu[L(\zeta')] \right] \tag{28}$$

$$= \sum_{i \in \{0,1\}} \frac{\tau_{ii}}{\mu[L(\zeta')]} \int_{\boldsymbol{x} \in L(\zeta')^c} h_i(\boldsymbol{x}) f(\boldsymbol{x}) d\boldsymbol{x} \left[ \int_{\boldsymbol{x} \in L(\zeta')} \frac{h_i(\boldsymbol{x})}{h_i(\zeta')} f(\boldsymbol{x}) d\boldsymbol{x} - \mu[L(\zeta')] \right] \tag{29}$$

Here let $I_i(\zeta') = \int\limits_{\boldsymbol{x} \in L(\zeta')^c} h_i(\boldsymbol{x}) f(\boldsymbol{x}) d\boldsymbol{x}$. Observe that

$$\int\limits_{\boldsymbol{x} \in L(\zeta')} \frac{h_i(\boldsymbol{x})}{h_i(\zeta')} f(\boldsymbol{x}) d\boldsymbol{x} = \int\limits_{\boldsymbol{x} \in L(\zeta)} \frac{h_i(\boldsymbol{x})}{h_i(\zeta')} f(\boldsymbol{x}) d\boldsymbol{x} + \int\limits_{\boldsymbol{x} \in L(\zeta') \backslash L(\zeta)} \frac{h_i(\boldsymbol{x})}{h_i(\zeta')} f(\boldsymbol{x}) d\boldsymbol{x} \qquad (30)$$

$$\geq \int\limits_{\boldsymbol{x} \in L(\zeta)} \left[ \frac{h_i(\boldsymbol{x})}{h_i(\zeta')} - 1 + 1 \right] f(\boldsymbol{x}) d\boldsymbol{x} + \int\limits_{\boldsymbol{x} \in L(\zeta') \backslash L(\zeta)} \frac{h_i(\zeta')}{h_i(\zeta')} f(\boldsymbol{x}) d\boldsymbol{x} \qquad (31)$$

$$\geq \int\limits_{\boldsymbol{x} \in L(\zeta)} \left[ \frac{h_i(\zeta) - h_i(\zeta')}{h_i(\zeta')} \right] f(\boldsymbol{x}) d\boldsymbol{x} + \mu[L(\zeta')] \qquad (32)$$

$$\geq \mu[L(\zeta)] \left[ \frac{h_i(\zeta) - h_i(\zeta')}{h_i(\zeta')} \right] + \mu[L(\zeta')] \qquad (33)$$

A valid choice of $\zeta$ implies $\mu[L(\zeta')] > 0$. Plug $I_i(\zeta')$ and (31) back into (27) and it end up with

$$\lim_{n \to \infty} (\ell'_{S_n, \mathcal{A}_\zeta} - \ell'_{S_n, \mathcal{A}_0}) \geq \sum_{i \in \{0,1\}} \tau_{ii} \frac{\mu[L(\zeta)]}{\mu[L(\zeta')]} \left[ \frac{h_i(\zeta) - h_i(\zeta')}{h_i(\zeta')} \right] I_i(\zeta') = C_\zeta > 0 \qquad (34)$$

Remember that $h_i(\zeta)$ is an increasing function and $\zeta > \zeta' = \frac{1}{2} \left( \zeta + \frac{1 + |\tau_{10} - \tau_{01}|}{2} \right)$. Then every term in (34) is positive and we end up with a positive constant $C_\zeta$.

$\square$

If there doesn't exists a point $\boldsymbol{x}$ such that $\eta_i(\boldsymbol{x}) = 0$, let $a_i = \min\limits_{\boldsymbol{x} \in \mathcal{X}} \eta_i(\boldsymbol{x})$ and $\widetilde{a}_i = (1 - \tau_{10} - \tau_{01}) a_i + \tau_{1-i,i}$, then the following generalized theorem applies for minimum purity.

**Theorem (General Minimum Purity Guarantee).** $\forall \delta > 0$, $\forall \zeta > \frac{1 + |\tau_{10} - \tau_{01}|}{2}$, there exist $N(\delta, \zeta) > 0$, $c_1(\zeta) > 0$, constant $c_2 \in \left( 0, \frac{e-1}{e} \right)$, and an non-decreasing function $g_1(\zeta) \in \left[ \min\limits_{i \in \{0,1\}} \frac{\tau_{ii} \tau_{1-i,i} (\zeta' - \widetilde{a}_i)}{(1 - \tau_{10} - \tau_{01}) \zeta' \widetilde{a}_i}, 1 \right]$, such that $\forall n \geq N(\delta, \zeta)$, $\forall q > 1$ and $\forall k \in [c_1(\zeta) \log^q n, c_2 n]$:

$$P \left[ (\ell_{S_n, \mathcal{A}_\zeta} - \ell_{S_n, \mathcal{A}_0}) > g_1(\zeta) \right] \geq 1 - \delta$$

**Remark:** Notice here $g_1(\zeta) > 0$, since $\widetilde{a}_i < \frac{1}{2} < \zeta'$. And for average purity the conclusion remains the same.

## A.2 Proof of the Abundancy Theorem

Denote $n_c = \#\{ \bigcup\limits_{i \in \{0,1\}} C^{(i)}(\zeta) \}$, where $C^{(i)}(\zeta)$ is data points of type $i$ that finally kept by our algorithm using parameter $\zeta$. We have:

**Theorem 2 (Abundancy).** $\forall \delta > 0$, $\forall \zeta > \frac{1 + |\tau_{10} - \tau_{01}|}{2}$, $\forall \epsilon > 0$, there exists $c_1(\zeta) > 0$, $c_2 \in (0, \frac{e-1}{e})$ and $N(\delta, \zeta, \epsilon) > 0$, such that $\forall n \geq N(\delta, \zeta, \epsilon)$, and $\forall k \in [c_1(\zeta) \log^q n, c_2 n]$, with probability at least $1 - \delta$:

$$\frac{n_c}{n} \geq \mu(L(\zeta))$$

*Proof.* Given Lemma 1, 2 and 3, $\forall \delta > 0$ and $\forall \zeta > \frac{1 + |\tau_{10} - \tau_{01}|}{2}$ then $\exists N(\delta, \zeta, \epsilon) > 0$ such that $\forall n > N(\delta, \zeta, \epsilon)$, $E_C, E_I$ and $E_F$ hold with probability at least $1 - \delta/4$. We know that $L(\zeta) \cap \mathbf{X} \subset \bigcup\limits_{i \in \{0,1\}} C^{(i)} \subset L(\zeta') \cap \mathbf{X}$. Thus for a set of i.i.d sampled points $\mathbf{X}$, $\exists \mu_\Delta \in (0, \mu(L(\zeta')) - \mu(L(\zeta)))$, $n_c = \sum\limits_{\boldsymbol{x} \in \mathbf{X}} b_{\boldsymbol{x}}$ and $b_{\boldsymbol{x}} \sim Bernoulli(\mu(L(\zeta)) + \mu_\Delta)$. Observe that $n_c$ stochastic dominates random variable $Binomial(n, \mu(L(\zeta)))$. So the MLE $\hat{\mu}(L(\zeta)) = \frac{n_c}{n} \xrightarrow{p} \mu(L(\zeta)) + \mu_\Delta \geq \mu(L(\zeta))$.

Let event $E_A = \{n_c/n \geq \mu(L(\zeta)) \mid E_C, E_I, E_F\}$, $\forall \delta > 0$ and $\forall \zeta > \frac{1+|\tau_{10} - \tau_{01}|}{2}$, $\exists N(\delta, \zeta, \epsilon) > 0$ such that $\forall n \geq N(\delta, \zeta, \epsilon)$, $P(E_A \mid E_C, E_I, E_F) \geq 1 - \delta/4$. As a result:

$$P(E_A \cap E_C \cap E_I \cap E_F) = P(\{E_A\} \cap \{E_F \mid E_C, E_I\} \cap \{E_I \mid E_C\} \cap \{E_C\})$$
$$\geq 1 - P(E_A^c) - P(E_F^c \mid E_I) - P(E_I^c \mid E_C) - P(E_C^c)$$
$$= 1 - 4 * (\delta/4) = 1 - \delta$$

In other words, $\forall \delta > 0$ and $\zeta > \frac{1+|\tau_{10} - \tau_{01}|}{2}$, $\exists N(\delta, \zeta, \epsilon) > 0$ such that if $n > N(\delta, \zeta, \epsilon)$ with probability at least $1 - \delta$, $\frac{n_c}{n} > \mu(L(\zeta))$.

$\square$

## B   On the Consistency with the Bayes Optimum

$\forall i \in \{0, 1\}$ and for the posterior probability $\eta_i(\boldsymbol{x})$, define $h_i^*(\boldsymbol{x}) = \delta_{\eta_i(\boldsymbol{x}) > \frac{1}{2}}(\boldsymbol{x})$. $h_i^*(\boldsymbol{x}) = 1$ indicates that $y(\boldsymbol{x}) = i$. The Bayes optimal classifier $h^*(\boldsymbol{x}) = \frac{1}{2}h_1^*(\boldsymbol{x}) + \frac{1}{2}[1 - h_0^*(\boldsymbol{x})]$.

In the paper, we provide theorems that lower bound the purity (consistency between noisy labels and true labels) of the final kept data points. However, like many existing works, we can also show the consistency between the label of final kept points given by our algorithm and the true Bayes optimal classifier, which is less challenging than guaranteeing the purity after connectivity, isolation and filtering (Lemmas 1, 2, and 3) are established. The following theorem will show that all labels of data points preserved by our algorithm will agree with Bayes optimal classifier's prediction with large probability.

**Theorem 3 (Consistency with $h^*(\boldsymbol{x})$).** $\forall \delta > 0$, $\forall \zeta > \frac{1+|\tau_{10} - \tau_{01}|}{2}$, *there exist constants* $N(\delta, \zeta) > 0$, $c_1(\zeta) > 0$, $c_2 \in \left(0, \frac{e-1}{e}\right)$, *such that* $\forall n \geq N(\delta, \zeta)$, $\forall q > 1$ *and* $\forall k \in [c_1(\zeta) \log^q n, c_2 n]$ *and* $\forall \boldsymbol{x} \in \bigcup\limits_{i \in \{0,1\}} C^{(i)}(\zeta)$:

$$P\left[\widetilde{y}(\boldsymbol{x}) = h^*(\boldsymbol{x})\right] \geq 1 - \delta \tag{35}$$

*Proof.* By combining Lemma 1, 2 and 3, $\forall \delta > 0$, $\forall \zeta > \frac{1+|\tau_{10} - \tau_{01}|}{2}$ and $\forall i \in \{0, 1\}$, $\exists N(\delta, \zeta) > 0$ such that $\forall n \geq N(\delta, \zeta)$, with probability at least $1 - \delta$, we have $[L(\zeta) \cap A_i^+ \cap \mathbf{X}] \subset C^{(i)}(\zeta) \subset [L(\zeta') \cap A_i^+ \cap \mathbf{X}]$. By definition, we know $\forall \boldsymbol{x} \in A_i^+$, we have $\eta_i(\boldsymbol{x}) > \frac{1}{2}$ then also $h_i^*(\boldsymbol{x}) = 1$, which implies that $h^*(\boldsymbol{x}) = i$. Since $\forall \boldsymbol{x} \in C^{(i)}$, $\widetilde{y}(\boldsymbol{x}) = i$, we have shown $\widetilde{y}(\boldsymbol{x}) = i = h^*(\boldsymbol{x})$. Then:

$$P[\widetilde{y}(\boldsymbol{x}) = h^*(\boldsymbol{x})] = P[\widetilde{y}(\boldsymbol{x}) = h^*(\boldsymbol{x}), E_C \cap E_I \cap E_F] + P[\widetilde{y}(\boldsymbol{x}) = h^*(\boldsymbol{x}), (E_C \cap E_I \cap E_F)^c] \tag{36}$$

$$\geq P[\widetilde{y}(\boldsymbol{x}) = h^*(\boldsymbol{x}) \mid E_C \cap E_I \cap E_F]P(E_C \cap E_I \cap E_F) \tag{37}$$

$$= P(E_C \cap E_I \cap E_F) \geq 1 - \delta \tag{38}$$

$\square$

## C   Discussion of Hyper-parameters in Our Method

### C.1   Analysis of Hyper-Parameter $\zeta$

One important hyper-parameter of our method is $\zeta$. In our algorithm, after the largest connected component of a class is chosen, we further filter it to ensure the purity. In particular, we consider a datum with label $\widetilde{y}$ clean only if at least a fraction of $\zeta$ of its $k$-nearest neighbors have the same label $\widetilde{y}$. The value of $\zeta$, ranging between 0 and 1, controls how selective we are in collecting clean data.

As suggested in Section 2.1 of the main paper, for a binary classification problem, $\zeta$ needs to be at least $1/2 + \epsilon$, for $\epsilon$ being an arbitrarily small positive constant. For a multiclass problem, the lower bound of $\zeta$ can be smaller than $1/2$. We may also consider taking a higher $\zeta$ in the beginning of the training to ensure the purity, and later relax to a lower $\zeta$ so that sufficient clean data are collected.

In practice, we observe that it suffices to take a constant $\zeta$ throughout the training. We also observe that the performance is very robust to the choice of $\zeta$. We show in Fig. 2 that for different datasets

with different noise patterns/levels, choosing $\zeta = 0.25$, $0.5$ and $0.75$ will all result in reasonably good performance. For all experiments reported in the main paper, *we simply set $\zeta = 0.5$.*

Figure 2: The effect of $\zeta$ on the model performance with different datasets/noise patterns. All the experimental settings are the same as the main paper.

## C.2   Other Hyper-Parameters

As discussed in the main paper, our method is very robust to (1) validation set size/cleanness; (2) $k_c$ in building $k$-nearest-neighbor graphs to compute the connected components; (3) $k_o$ in computing $k$-nearest neighbors for $\zeta$-filtering; and (4) feature space dimension (dimension of the corresponding neural network layer). In Fig. 4 of the main paper, we already provided results on CIFAR-10, 60% uniform noise. Below we provide similar results on other datasets/noise settings. As is shown, our method is robust to the size and purity of validation set. Besides, it is not sensitive to the feature dimensions and the $k$ in computing nearest neighbors. In our experiments, we choose a clean validation set of size 10k for model selection. We set $k_c = 4$, $k_o = 32$ and feature dimension$= 512$.

Figure 3: Hyper-parameter analysis for CIFAR-100, 60% uniform noise: (a) validation set; (b) $k_c$; (c) $k_o$; (d) Feature dimension. For each figure, we change one of the parameters while keeping the others fixed (to $k_c = 4$, $k_o = 32$, feature dimension = 512, validation set = clean 10k).

Figure 4: Hyper-parameter analysis for CIFAR-10, 30% pair-flipping noise: (a) validation set; (b) $k_c$; (c) $k_o$; (d) Feature dimension. The parameter specifications are the same with those in Fig. 3 above.

Figure 5: Hyper-parameter analysis for CIFAR-100, 30% pair-flipping noise: (a) validation set; (b) $k_c$; (c) $k_o$; (d) Feature dimension. The parameter specifications are the same with those in Fig. 3.

## C.3    Additional Experiments on the Point Cloud Data Domain

To test the applicability of our method to the data beyond image domains, we also conduct experiments on the point cloud data. Specifically, we adopt the ModelNet40 [4] dataset, which contains 12,311 CAD models from 40 categories, with 9,843 used for training and 2,468 for testing. In the experiment, we split 20% from the training set as validation data. The CAD models are organized in triangular meshes, and we follow the protocol of [3] to convert them into point clouds by uniformly sampling 1,024 points from the mesh and normalizing them within a unit ball. We employ PointNet [3] for point cloud classification. The results are shown in Table 1. We observe similar advantages of our method over the baselines on the point cloud dataset.

Table 1: Comparison of test accuracies on ModelNet40 under different noise types and fractions. The average accuracies and standard deviations over 5 trials are reported.

| Method | Uniform Flipping | | Pair Flipping | |
|---|---|---|---|---|
| | 40% | 80% | 20% | 40% |
| Standard | $74.7 \pm 1.2$ | $56.0 \pm 2.1$ | $83.4 \pm 1.1$ | $77.5 \pm 2.1$ |
| Forgetting | $74.7 \pm 1.2$ | $56.8 \pm 2.9$ | $83.4 \pm 1.1$ | $77.4 \pm 2.0$ |
| Bootstrap | $75.6 \pm 2.8$ | $57.1 \pm 3.1$ | $84.5 \pm 0.5$ | $60.8 \pm 4.3$ |
| Forward | $41.7 \pm 5.2$ | $19.8 \pm 4.8$ | $52.0 \pm 2.0$ | $51.3 \pm 5.5$ |
| Decoupling | $79.2 \pm 1.0$ | $54.6 \pm 2.8$ | $85.9 \pm 0.2$ | $69.2 \pm 2.3$ |
| MentorNet | $74.9 \pm 2.6$ | $56.2 \pm 1.8$ | $83.8 \pm 1.1$ | $69.9 \pm 2.6$ |
| Co-teaching | $82.8 \pm 1.1$ | $69.3 \pm 3.2$ | $84.5 \pm 0.5$ | $77.6 \pm 1.9$ |
| Co-teaching+ | $83.0 \pm 1.2$ | $62.2 \pm 9.2$ | $85.0 \pm 0.9$ | $73.9 \pm 4.2$ |
| IterNLD | $75.3 \pm 0.9$ | $55.7 \pm 2.3$ | $84.2 \pm 1.3$ | $76.9 \pm 2.2$ |
| RoG | $80.0 \pm 0.9$ | $43.5 \pm 3.2$ | $81.5 \pm 1.1$ | $76.2 \pm 0.9$ |
| PENCIL | $81.2 \pm 1.1$ | $61.7 \pm 2.1$ | $84.8 \pm 0.4$ | $78.7 \pm 1.4$ |
| GCE | $83.1 \pm 0.5$ | $63.0 \pm 5.6$ | $83.4 \pm 0.7$ | $68.7 \pm 2.6$ |
| SL | $78.8 \pm 0.5$ | $59.3 \pm 1.9$ | $81.5 \pm 0.8$ | $68.8 \pm 1.3$ |
| TopoFilter | $\mathbf{84.2 \pm 0.6}$ | $\mathbf{70.4 \pm 2.6}$ | $\mathbf{86.4 \pm 0.4}$ | $\mathbf{79.6 \pm 1.4}$ |

## Footnotes

[1]In the case when this minimum is not 0 but some value $a \in (0, 1)$, the expressions become more unwieldy, and we derive it in general minimum purity guarantee theorem in this subsection later. Our assertion for average purity holds regardless of this condition.