[Reviews · NeurIPS 2020]

Review 1

Summary and Contributions: === after author's rebuttal == I have read the author response and thank them for answering my questions on the relation to Topological Data Analysis and clarifying the topological (as opposed to geometric) aspect of the paper. I lean more towards accepting this paper. =============== This work studies the problem of handing label noise - a realistic problem that concerns practitioners in a variety of applications. The proposed solution uses inherent geometric structure induced by nearest neighbors to discover large correlations where connected components are preserved. Motivated by further corruption, an iterative algorithm is devised that peels layers off connected components based on a parameter \gamma. Theoretical results are then provided to negate any ambivalence against this method disrupting the purity or abundance of the dataset.

Strengths: The major strength of this work is an algorithm that allows deep learning practitioners to couple with a way to handle label noise. This yields quite a modular pathway for future applications who are interested in robustification against label noise. Section 2 does a great job with regards to presentation in that, the entire algorithm can be understood from this section. A noise filtering algorithm is always faced with two challenges: (1) ensuring that the algorithm recognizes proper noise and (2) ensuring the algorithm is not too conservative in the sense that the distribution is disrupted. The authors acknowledge this precisely in the form of 'purity' and 'abundance' respectively and present novel theoretical results. Although the results in their current are difficult to digest or interpret, it is important to take as step in this direction. The significance of the work is related to the problem they solve, which has very clear application. The proposed method easily finds a place in the NeurIPS community and and within practicing machine learning at large.

Weaknesses: The 'purity' and 'abundance' guarantee is important as mentioned above however the current form of the results are difficult to digest. While I understand this is a challenge, it would be more useful to the audience if the presentation was more improved at the beginning, which is in contrast to the strength of Section 2. For example on Line 151, partitions A_{i}^+ and A_{i}^- are introduced however their use and explanations are exposed later. It would improve presentation if Figure 2 along with intuition is provided to guide the reader on what these sets mean in reference to the algorithm. Another limitation is that this paper is specifically derived for the Euclidean domain. While this is a reasonable limitation, it would be more assuring to know how fixated these results in the Euclidean space and how they behave in non-euclidean domains with arbitrary distance measures.

Correctness: The claims and empirical methodology are correct.

Clarity: The clarity is well-written on Section 2, which describes the algorithm using rather familiar and self-contained notation. On the other hand, the Theoretical results in Section 3 can be difficult to follow or understand.

Relation to Prior Work: I am not an expert in this area however I believe majority of the important existing work are discussed.

Reproducibility: Yes

Additional Feedback: Questions: (1) I question the use of naming this work based on "topological" structure since I believe it uses more geometric structure, especially since it seems to assume a Euclidean space. A more topological approach would consider connected components in the sense of holes based on persistence diagrams and homological properties of the dataset as pointclouds. This is typical in the are of Topological Data Analysis (TDA). What sense is the proposed work topological? (more than it is geometric?) (2) Is there any intuition as to how this work would scale to generalization to other spaces that are not Euclidean? The idea is intuitive however I am not sure if the guarantees are easily transferable.


Review 2

Summary and Contributions: The authors propose a technique to handle label noise in supervised learning problems; in particular, by looking at the spatial, latent feature distribution (obtained by a deep learning model) the proposed technique is able to isolate outliers and filter out noisy labels in multi-class training data. The idea seems novel, theoretical justification is provided and experimental observations are encouraging.

Strengths: The problem is quite significant, proposed idea seems intuitive and sound, applies to practical ML training scenarios (deep learning architectures) and problem settings, and experimental results (though not comprehensive) are promising.

Weaknesses: The paper could be improved on a couple of aspects: (1) the proposed approach is generic, and I don't see a reason for it to be tied to deep neural architectures (though, admittedly, it's the most interesting practical scenario); I wonder if the writing can bring this out more explicitly. More importantly, then, could this approach also filter out label noise by working with the ambient feature space when training other, simpler, linear models or trees/forests. (2) Entire discussion and guarantees in Section 2.1 is meant for some feature space with the stated assumptions on label noise, class-conditional distributions, etc. But then the Algorithm is iterative, where the features get continuously refined. The paper seems to ignore this aspect; can something be said about purity or abundancy improving with iterations, which is observed in experiments? Perhaps not for a deep learning model, but maybe for a simple linear model to begin with.

Correctness: Yes, for the most part. There's one issue I had with the main theorem statements, I believe it's minor. a) Some normalization factor seems to be missing in the guarantees of Theorem 1; in particular, in the trivial case when the entire data is clean (the purity of A_0 is therefore 1), but in this setting even g(.) also seems to reduce to 1, which doesn't make sense because the purity is at most 1. So, shouldn't g(.) + purity (A_o) <= 1? b) Also, in Theorem 2, \mu seems to be undefined? ---- Thanks for clarifying these in the rebuttal. Please update the paper to reflect these. ----

Clarity: Yes, for the most part.

Relation to Prior Work: I'm not fully aware of deep learning + label noise literature, but the references & discussion seem adequate.

Reproducibility: Yes

Additional Feedback: I have read the author responses to my and other reviewers' concerns. I'm leaning more towards accepting the paper.


Review 3

Summary and Contributions: The authors present a method to improve performance in the presence of label noise that leverages the spatial behaviour of the latent feature representation to identify the clean data. Theoretical and empirical results are then given.

Strengths: I admire the authors attempt to ground an algorithm with theoretical foundations which is then investigated with experiments on real-world data sets and good improvements are uncovered.

Weaknesses: Theorem 1 appears to be stated in quite an unusual way, the details of which I will clarify. I can't see how the bounds depend on the variables k or δ. Usually it would be the case that a bound of this form is presented in a way so that strength of the gap is dependant on \delta. That is, of the form: Pr{ X - Y > f(δ) } ≥ 1 - δ, where f : R → R is decreasing. That way when we require the bound hold with more and more certainty, (decrease δ), the strength of the claim (the size of the gap X - Y) is qualified. As Theorem 1 is currently written, the claim is not very powerful, since the numbers g(ζ) and C_ζ are not bounded with respect to δ. For example, it might be the case that for every 0 ≤ δ ≤ 1 we have C_ζ positive and very close to 0, in which case Theorem 1 (2) tells us very little. Secondly regarding Theorem 1 (1), consider the case in which the transition matrix is the identity operator. τ_ii = 1, τ_ij = 0 for i≠j. That is, zero label noise. Then ζ ≤ g(ζ) ≤ 1, and we can arbitrarily increase the minimum purity gap by inflating ζ. Does this seem sensible? I would have appreciated a study of the time-complexity of the algorithm. Is the algorithm performant enough for large data sets? --- Update --- I wish to thank the authors for clarifying some of the issues above I raised. I still find Theorem 1 quite difficult to interpret.

Correctness: The authors display results and standard deviations from running the algorithm five times on CIFAR-10 and CIFAR-100. While the average performance is highlighted as superior to a number of appropriate baseline measures, the authors do not compute any statistics to conclude that TopoFilter is superior. In a number of cases the standard deviations are large and overlap with other state-of-the-art performance metrics, causing me to think that the superiority conclusion is not robust. The experimental section would benefit from more extensive testing of their algorithm and statistical analysis of the performance. --- Update --- The experimental results are encouraging. I would appreciate seeing the t-test results included in the paper.

Clarity: The enthusiastic use of calligraphic math font makes it a bit difficult to keep track of the notation. For example, \mathcal{C} : number of classes \mathcal{X} : Euclidean vector space \mathcal{Y} : discrete label set \mathcal{F} : joint distribution over \mathcal{X}\times\mathcal{Y} \mathcal{A} : purity concept It is more typical to reserve similar looking notation for similar types of mathematical objects. I believe the exposition would be improved by making the choice of math font more systematic. The indices of \tau and \eta should follow the set \mathcal{Y} = {1, 2, \dots }, but instead index from the set {0, 1}. It would be more clear and consistent to either renumber \mathcal{Y} or renumber \tau and \eta. L#158: The average purity concept appears to be missing a set cardinality operation |C_i|.

Relation to Prior Work: The authors provide a good overview of the literature on this topic.

Reproducibility: Yes

Additional Feedback: The broader impact section should expound on how the work relates to broader impact on society, not just the field of machine learning, which should differentiate it from the conclusion.


Review 4

Summary and Contributions: =============== Review Updates I appreciate the detailed responses from the authors about my concerns. Most of my questions are, in my point of view, properly answered. I updated the overall score. =============== This study presents the novelty way of dealing with label noise in data. To find noisy data, topological information in the latent representation space is exploited where clean and noise data can be well separated. And theoretical proofs for the proposed methods make them consolidate. The empirical improvement in performance is shown by comparing to other previous works for CIFAR 10/100 and Clothing1M dataset.

Strengths: The biggest strength of the paper is its effectiveness in plenty of mathematical proofs. All the claims of the paper are proved by statistical learning theory, which makes the method logical and consolidated. And it opens the door to the possibility for wide applications of the proposed approach by the case study.

Weaknesses: - This paper presents two flipping method for generating corrupted labels: Uniform and Pair. However, it is confusing that the method from previous works uses a different flipping method such as 'Symmetric' and 'Asymmetric' from [28], [39], or [43]. And they are all compared with the method of Uniform and Pair in this article, which are not in previous works. It is curious why the two flipping is proposed and used for comparison to previous works. - Though the proposed method is compared to many previous works, the number of the dataset used is limited(two dataset CIFAR 10/100 from similar domain and Clothing1M). Authors need to consider giving performance comparison to other datasets so that the excellence of the method be stood out.

Correctness: All the claims and methods are proved by the theoretical background. And the proposed method is applied to empirical studies for validating its effectiveness.

Clarity: Overall, this paper is well organized and logical. But one thing to comment is that it would be better if there are more explanations for the algorithm. Figure 2 is well illustrated. But the figure itself may be insufficient to clearly convey the complex idea to readers. Authors may consider to present the idea in detail with more figures or flow charts.

Relation to Prior Work: The study on dealing with noisy labels has a long history from the statistical learning field. By presenting related works and doing performance comparison to them, it is clear that the excellence of the proposals will be stood out.

Reproducibility: No

Additional Feedback: - Line 48-49 in Introduction : It is unclear to understand the meaning of the sentence. Is '.' right instead of ','? - Line 97 in Method : Authors may consider making 'early-stopped' more specific in the algorithm or equations. - As the proposed approach is composed of several procedures, there are some hyperparameters to be determined such as m or k. Is there any rule for finding proper value in a new problem? It may be helpful for practitioners who want to use the method.

[Author Response · NeurIPS 2020]

We thank the reviewers for their constructive feedback. We will improve the presentation according to the suggestions. Below we address some major concerns.

**Q1 [R1]: Does this work generalize to non-Euclidean domains with arbitrary distance measures?** Yes, both the algorithm and the theorems (with appropriate bounds depending on the metric) generalize to any metric space.

**Q2 [R1]: In terms of the name, the proposed work is more "geometric" than "topological". Relation to TDA.** Indeed, a persistent-homology-based loss was the first thing we tried! Unfortunately, the noisy classifier as a filter function was not very helpful. We ended up using the label as the filter function and the algorithm evolved into its current form. While we agree that geometry is very important, our guarantees rely on connectivity, and hence topology is important too. A geometry-relevant filter function for persistence is a candidate worth exploring in the future.

**Q3 [R2]: What is $\mu$ in Theorem 2 (Abundancy)?** We think you are referring to Theorem 5. $\mu(L(\zeta))$ is the probability measure of $L(\zeta)$, the $\zeta$-superlevel set of $\eta$. In general $\mu(A) = \int_A f$, where $f$ is the density.

**Q4 [R2]: Should be explicit the method is generic. Generalize to the ambient feature space?** Thanks for pointing this out. We will make this explicit. Yes, our method works in the ambient space with similar guarantees on purity. However, the abundancy could be limited; the largest component for label $i$ will still be pure, but could be small as it only covers one (the largest) piece of the true region of label $i$ (in which $\eta_i \geq 1/2$). In the ambient space, the true region of label $i$ can be scattered and even the largest piece can be small. In a deep layer of a neural net, this is not an issue, as the network is pulling the true regions of label $i$ together.

**Q5 [R2]: Can something be said about purity or abundancy improved with iterations if the algorithm?** The purity remains high throughout the iterations. The abundancy grows gradually through the iterative algorithm (as shown in the brown and blue curves in Fig. 3). As stated in Q4, the abundancy depends on the size of the largest true region of label $i$. This region will grow gradually as the network improves. An iterative version requires decreasing $\zeta$ carefully to 0.5 to get convergence and collect most of the pure data. We leave this for future work.

**Q6 [R2, R3]: Re. Theorem 1, when the data is clean, $g(\zeta)$ reduces to 1.** Thanks for pointing this out. If the original data is clean, indeed our algorithm does not (cannot) gain purity (either minimum or average). However, all our theorems hold as stated as long as the noise is not zero. This is because a) by definition the minimum purity of any impure dataset is 0 (see line 165 equation (15) in supplementary), and part 1 of the theorem applies, and b) the $C_\zeta$ in part 2 of the theorem will decrease to zero as noise goes to zero. As long as the noise is not zero, choosing a higher $\zeta$ will give us higher gains ($g(\zeta)$ and $C_\zeta$, respectively) in the purities.

**Q7 [R3]: It is unclear how the bounds depend on the variables $k$ or $\delta$.** In our theorem, the $\forall \delta, \zeta, q$ quantifiers come *before* the $\exists N, C_1, C_0$ quantifiers, and so $N$ itself depends on $\zeta$ and $\delta$. Since $n \geq N$, the left hand side does depend on $\delta$. We will write $N(\delta, \zeta, q)$, $C_1(\delta, \zeta, q)$ and $C_0(\delta, \zeta, q)$, to make this explicit. We will remove the potentially misleading word "constants" from the statement, which was put to indicate that they are independent of the distribution.

**Q8 [R3]: I would have appreciated a study of the time-complexity of the algorithm.** The exact construction of KNN takes $O(n^2)$ time (up to $poly(k, d)$ factors), but using $c$-approximate nearest neighbor methods would reduce the exponent to $(1 + 1/c^2)$. In practice, we use GPU implementation to speed up the construction. The rest (computing connected components using BFS, $\zeta$-filtering) takes $O(kn)$. Each iteration takes about 1 second for CIFAR10 (line 233-235) and 25 seconds for clothing 1M (line 274).

**Q9 [R3]: Would be better to provide more testing of the algorithm and statistical analysis of the performance.** Thanks for the suggestion. We also performed unpaired $t$-test (95% significance level) on the difference between the testing accuracy on CIFAR10/100. The improvement due to our method over state-of-the-art methods is statistically significant for all noise settings. Note that for Clothing1M, since the results of baseline methods in Table 2 are copied from published works (which did not provide standard deviation), we did not perform the $t$-test.

**Q10 [R4]: Uniform and pair flippings are different from symmetric and asymmetric flippings.** The uniform/pair flippings are exactly the same as symmetric/asymmetric flippings in [28, 39, 43]. We call it uniform/pair to emphasize the noise generation procedure, following [13, 18].

**Q11 [R4]: The number of datasets used is limited (CIFAR 10/100 and Clothing1M).** The CIFAR10/100 and Clothing1M have been widely adopted as the testbeds for studying algorithms robust to label noise. Following your suggestion, we also conduct experiments on ModelNet40 [1], which contains CAD models from 40 categories. We convert the CAD models into point clouds according to [2], and employ PointNet [2] for point cloud classification. This dataset offers a different domain from images. Due to the space, here we only list the results of some representative baselines under 0.4 uniform noise, and more results will be included in the final version.

| Standard | Co-teaching | Co-teaching+ | RoG | PENCIL | GCE | SL | TopoFilter |
|---|---|---|---|---|---|---|---|
| $74.7 \pm 1.2$ | $82.8 \pm 1.1$ | $83.0 \pm 1.2$ | $80.0 \pm 0.9$ | $81.2 \pm 1.1$ | $83.1 \pm 0.5$ | $78.8 \pm 0.5$ | $\mathbf{84.2 \pm 0.6}$ |

[1] Z. Wu, *et al*. 3D ShapeNets: A Deep Representation for Volumetric Shapes. CVPR, 2015.

[2] C. R. Qi, *et al*. PointNet: Deep Learning on Point Sets for 3D Classification and Segmentation. CVPR, 2017.

**Q12 [R4]: How to choose $m$ and $k$?** We can use the validation set to choose $m$ and $k$. When there is no validation set, we choose the burn-in milestone $m$ as the epoch when the training accuracy's increasing trend slows down. As for $k$, our ablation study (Fig. (4) in the main paper and Fig. (2-4) in the supplementary) shows it is quite robust. Any value between 4 and 64 gives a good performance. This is consistent with the $k \in [\Omega(\log n), O(n)]$ bound in Thm 1.

[Meta-Review · NeurIPS 2020]

All reviewers had a positive overall impression of this paper, and highlighted some salient features of the work: + generic solution to the important problem of coping with label noise + interesting, somewhat novel approach backed by theoretical guarantees + encouraging empirical results In terms of weaknesses, it was pointed out that the empirical comparisons are only done on three datasets (CIFAR-10, 100, and MNIST), and that the discussion of Theorem 1 could be improved. The former appears reasonable for a novel idea that with a core theoretical contribution, although the authors are encouraged to incorporate elements of their response in updating the discussion around Theorem 1.